# Efficient Fairness-Performance Pareto Front Computation

Mark Kozdoba [*1], Binyamin Perets [*1], and Shie Mannor[1,2]

[1]Technion - Israel Institute of Technology, Haifa, Israel
[2]NVIDIA

## Abstract

There is a well known intrinsic trade-off between the fairness of a representation and the performance of classifiers derived from the representation. In this paper we propose a new method to compute the optimal Pareto front of this trade off. In contrast to the existing methods, this approach does not require the training of complex fair representation models.

Our approach is derived through three main steps: We analyze fair representations theoretically, and derive several structural properties of optimal representations. We then show that these properties enable a reduction of the computation of the Pareto Front to a compact discrete problem. Finally, we show that these compact approximating problems can be efficiently solved via off-the shelf concave-convex programming methods.

In addition to representations, we show that the new methods may also be used to directly compute the Pareto front of fair classification problems. Moreover, the proposed methods may be used with any concave performance measure. This is in contrast to the existing reduction approaches, developed recently in fair classification, which rely explicitly on the structure of the non-differentiable accuracy measure, and are thus unlikely to be extendable.

The approach was evaluated on several real world benchmark datasets and compares favorably to a number of recent state of the art fair representation and classification methods.

## 1 Introduction

Fair representations are a central topic in the field of Fair Machine Learning, Mehrabi et al. (2021), Pessach and Shmueli (2022),Chouldechova and Roth (2018). Since their introduction in Zemel et al. (2013), Fair representations have been extensively studied, giving rise to a variety of approaches based on a wide range of modern machine learning methods, such GANs, variational auto encoders, numerous variants of Optimal Transport methods, and direct variational formulations. See the papers Feldman et al. (2015), Madras et al. (2018), Gordaliza et al. (2019); Zehlike et al. (2020), Song et al. (2019), Du et al. (2020), Zhao and Gordon (2022), Jovanović et al. (2023), Dehdashtian et al. (2024), for a sample of existing methods.

For a given representation learning problem and a target classification problem, since the fairness constraints reduce the space of feasible classifiers, the best possible classification performance will usually be lower as the fairness constraint becomes stronger. This phenomenon is known as the Fairness-Performance trade-off. Assume that we have fixed a way to measure fairness. Then for a given representation learning method, one is often interested in the fairness-performance curve $(\gamma, E(\gamma))$. Here, $\gamma$ is the fairness level, and $E(\gamma)$ is the classification performance of the method at

39th Conference on Neural Information Processing Systems (NeurIPS 2025).

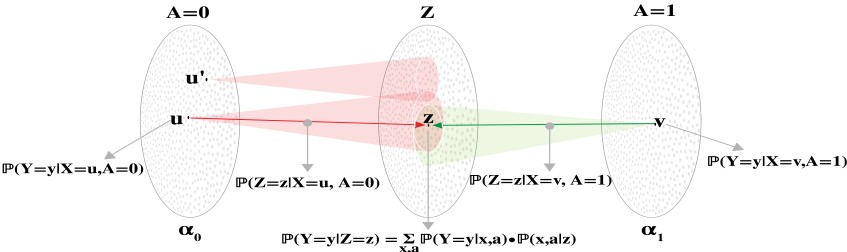

Figure 1: Fair Representation Problem Setting.

that level. The curve $(\gamma, E(\gamma))$ where $E(\gamma)$ is the best possible performance over all representations and classifiers under the constraint is known as the Fairness-Performance Pareto Front.

As indicated by the above discussion, representation learning methods typically involve models with high dimensional parameter spaces, and complex, possibly constrained non-convex optimisation algorithms. As such, these methods may be prone to local minima and sensitivity to a variety of hyper-parameters, such as architecture details, learning rates, and even initializations. While the representations produced by such methods may often be useful, it nevertheless may be difficult to decide whether their associated Fairness-Performance curve is close to the true Pareto Front.

In this paper we propose a new method to compute the optimal Pareto front, which does not require the training of complex fair representation models. In other words, we show that, perhaps somewhat surprisingly, the computation of the Pareto Front can be decoupled from that of the representation, and only relies on learning of much simpler, unconstrained classifiers on the data. To achieve this, we first show that the optimal fair representations satisfy a number of structural properties. While these properties may be of independent interest, here we use them to express the points on the Pareto Front as the solutions of small discrete optimisation problems. These problems, known as *concave minimisation* problems Benson (1995), have been extensively studied and can be efficiently solved using modern dedicated optimisation frameworks, Shen et al. (2016).

We now describe the results in more detail. Let $X \in \mathcal{X}$ and $A \in \mathcal{A}$ denote the data features and the sensitive attribute, respectively. We assume that $A$ is binary, while $X$ may take values in an arbitrary space $\mathcal{X}$, typically with $\mathcal{X} = \mathbb{R}^d$. In addition, we have a *target* variable $Y$, taking values in a finite set $\mathcal{Y}$. We are then interested in representations that maximise the performance of prediction of $Y$, under the fairness constraint. The representation is denoted by $Z$, and is typically expressed by constructing the conditional distributions $\mathbb{P}(Z|X = x, A = a)$, for $a \in \{0, 1\}$. The problem setting is illustrated in Figure 1.

Our approach consists of three main steps: We first observe that one can map the data features $(x, a) \in \mathcal{X} \times \mathcal{A}$ to a much smaller space $\Delta_{\mathcal{Y}}$ of distributions on the set of label values $\mathcal{Y}$, without loosing any information necessary for the computation of the Pareto front. The mapping is done by means of the optimal Bayes classifier. This result is referred to as Factorization Lemma, Section 3.2, where the mapping is done via the optimal Bayes classifier. Similar arguments were recently implicitly used in the study of fair classification tradeoffs, (Xian et al., 2023; Wang et al., 2023), but were restricted to classification and to accuracy loss (Section 2).

The advantage of working with a small space such as $\Delta_{\mathcal{Y}}$ is that it can be easily discretized. For instance, if $Y = 0, 1$, then we essentially have $\Delta_{\mathcal{Y}} = [0, 1]$, which is descretised trivially. We note that other, data dependent discretisation schemes, such as clustering, maybe possible for problems involving highly multi label targets. Alternatively, one can also consider the dataset itself as a grid, a view that is typically taken by transportation based approaches, e.x. Gordaliza et al. (2019), Xian et al. (2023).

Next, assuming the data is discretized and finite, we ask how large the represntation space $\mathcal{Z}$ should be, in order to support both optimal performance and fairness? For instance, we believe the answer to the following question is not apriori obvious: Can representations on infinite spaces be approximated, in terms of performance and fairness, by representations on finite and bounded spaces $\mathcal{Z}$? These questions are addressed by the Invertibility Theorem, Section 3.3, which asserts that all optimal representations may be taken in a certain canonical form, which we term *invertible*. This

result, in conjunction with an additional approximation lemma, is used in Section 4.1 to construct representations with any desired degree of approximation.

Finally, based on these results in Section 4.1 we also introduce the MIFPO (Model Independent Fairness-Performance Optimization), a discrete optimisation problem that is essentially equivalent to a computation of the fairness-performance tradeoff on a discrete set. We show that in this situation MIFPO is a concave minimisation problem with linear constraints, and we solve it using the disciplined convex-concave programming framework, DCCP, Shen et al. (2016).

We evaluate our approach on standard fairness benchmark datasets and compare its fairness-performance curve to multiple state-of-the-art fair representation methods. We also compare MIFPO to the fairness-performance Pareto front of fair *classifiers*[1]. As expected, MIFPO effectively serves as an upper bound on almost all other algorithms in both cases.

To summarise, the contributions of this paper are as follows: **(a)** We derive several new structural properties of optimal fair representations. **(b)** We use these properties to construct a model independent problem, MIFPO, which can approximate the Pareto Front of arbitrary high dimensional data distributions, but is much simpler to solve than direct representation learning for such distributions. **(c)** We illustrate the approach on real world fairness benchmarks.

The rest of this paper is organised as follows: Section 2 discusses the literature and related work. In Section 3 we discuss the theoretical results, including factorization and the Invertibility Theorem. The MIFPO problem construction and the full Pareto Front computation algorithm are provided in Section 4. Experimental results are presented in Section 5, and we conclude the paper in Section 6. All proofs are provided in the Supplementary Material.

## 2  Literature and Prior Work

We refer to the book Barocas et al. (2023), and surveys Mehrabi et al. (2021),Du et al. (2020), for a general overview of representations. Tradeoffs in particular where explicitly studied in Song et al. (2019), Balunović et al. (2022a), Zhao and Gordon (2022), Jovanović et al. (2023),Dehdashtian et al. (2024), among others.

In this paper we use the total variation based fairness constraints, similarly to the line of work in Madras et al. (2018), Zhao and Gordon (2022), Balunović et al. (2022a), Jovanović et al. (2023). Other constraints used in the literature include entropy based constraints, Song et al. (2019), or RKHS based independence test constraints, Dehdashtian et al. (2024).

As discussed earlier, the vast majority of the work above concentrates on finding neural network based fair representations via involved optimization schemes with possible local minima, which may be hard to analyze. This highlights the usefulness of our approach direct approach to the computation of the Pareto front, which has clear theoretical grounding, and in which sources of approximation error are well understood and may be controlled.

Relations between fairness and performance were studied in Zhao and Gordon (2022). In particular, for *perfectly fair* representations, they derived lower bounds on the accuracy in terms of the difference of the base rates between the groups. However, this work did not introduce new algorithms for the computation of fair representations or of the associated Pareto front. The extension of the considerations in this paper to the full front was carried in Xian et al. (2023) for classification (see below).

The accuracy fairness tradeoff has also been extensively studied in the context of fair classification (without representations), see for instance Agarwal et al. (2018), Kim et al. (2020) , Alghamdi et al. (2022), Xian et al. (2023), Wang et al. (2023), for a sample of recent approaches. In particular, the papers Xian et al. (2023), Wang et al. (2023) are state of the art, and are also the most closely related to our methods, among the existing work.

Similarly to our approach, the analysis in these two papers starts with the estimation of the probabilities $\mathbb{P}(Y|X, A)$, which are then used to compute the constrained performance. However, the subsequent steps are different. Crucially, the analysis in both Xian et al. (2023) and Wang et al. (2023) relies critically on the properties of the accuracy as the performance metric. Consequently,

---

[1]See Sections 2 and 3.2 for the relation between our representation framework and fair classification results.

it can not be extended to general concave performance measures, such as the standard (minus) log loss, for instance. Roughly speaking, in the appropriate sense, accuracy largely ignores classification probabilities. This allows the simple description of classifiers as small confusion matrices in Wang et al. (2023) (extending the approach of Kim et al. (2020)), and the restriction of the distributions to the vertices of the simplex in Xian et al. (2023). The special structure of accuracy is highlighted also in our Lemma 3.1, where we show that classifiers with accuracy may be effectively described by representations using only 2 points. We conclude that even when restricted to classification, our approach analyses a fundamentally more complex situation compared to previous work. On the other hand, Xian et al. (2023) and Wang et al. (2023) support non binary sensitive attributes and group labels, while such an extension for our methods is out of scope for this paper. A comparison of computational complexities for these algorithms may be found in Supplementary J.

## 3 Structure of Fair Representations

In this Section we describe several theoretical properties of fair representations. In Section 3.1 we introduce the problem setup and the necessary notation. In Section 3.2 we discuss relations to classification with accuracy loss and the factorization result, which allows to reduce the size of the representation space. The Invertibility Theorem is introduced in Section 3.3.

### 3.1 Problem Setting

Let $A$ be a binary sensitive variable, and let $X$ be an additional feature random variable, with values in a set $\mathcal{X}$, typically with $\mathcal{X} = \mathbb{R}^d$. Assume also that there is a target variable $Y$ with finitely many values in a set $\mathcal{Y}$, jointly distributed with $X, A$.

A representation $Z$ of $(X, A)$ is defined as a random variable taking values in some space $\mathcal{Z}$, with *(i)* distribution given through $\mathbb{P}_\theta (Z|X, A)$, where $\theta$ are *the parameters of the representation*, and *(ii)* such that conditioned on $(X, A)$, $Z$ is independent of the rest of the variables of the problem. In particular, we have

$$Z \perp\!\!\!\perp Y \ |(X, A), \tag{1}$$

where $\perp\!\!\!\perp$ denotes statistical independence.

Fairness in this paper will be measured by the Total Variation distance. For two distributions, $\mu, \nu$ on $\mathbb{R}^d$, with densities $f_\mu, f_\nu$, respectively, this distance is defined as

$$\|\mu - \nu\|_{TV} = \frac{1}{2} \sup_{g \text{ s.t. } \|g\|_\infty \leq 1} \int g(x) \cdot [f_\mu(x) - f_\nu(x)] \, dx \quad = \frac{1}{2} \int |f_\mu(x) - f_\nu(x)| \, dx. \tag{2}$$

Note that $\int |f_\mu(x) - f_\nu(x)| \, dx$ is in fact the $L_1$ distance, and the equivalence $\|\cdot\|_{TV} = \frac{1}{2} \|\cdot\|_{L_1}$ is well known, see Cover and Thomas (2012).

For $a \in \{0, 1\}$, let $\mu_a$ be the distribution of $Z$ given $A = a$, i.e. $\mu_a(\cdot) := \mathbb{P}(Z = \cdot | A = a)$. We denote the distance induced by the representation as $D_{TV}(Z) = \|\mu_0 - \mu_1\|_{TV}$, and for $\gamma \geq 0$, we say that the representation $Z$ is $\gamma$-fair iff

$$D_{TV}(Z) = \|\mu_0 - \mu_1\|_{TV} \leq \gamma \qquad \text{(Fairness Condition)}. \tag{3}$$

Note that (3) is a quantitative relaxation of the "perfect fairness" condition in the sense of statistical parity, which requires $Z \perp\!\!\!\perp A$. Specifically, observe that by definition, $Z \perp\!\!\!\perp A$ iff (3) holds with $\gamma = 0$ (i.e. $\mu_0 = \mu_1$). In addition, as shown in Madras et al. (2018), (3) implies several other common fairness criteria, in particular, bounds on demographic parity and equalized odds metrics for any downstream classifier built on top of $Z$.

Next, we describe the measurement of information loss in $Y$ due to the representation. Let $h : \Delta_{\mathcal{Y}} \to \mathbb{R}$ be a continuous and concave function on the set of probability distributions on $\mathcal{Y}$, $\Delta_{\mathcal{Y}}$. The quantity $h(\mathbb{P}(Y|X = x))$ will measure the best possible prediction accuracy of $Y$ conditioned on $X = x$, for varying $x$. As an example, consider the case of binary $Y$, $\mathcal{Y} = \{0, 1\}$. Every point in $\Delta_{\mathcal{Y}}$ can be written as $(p, 1 - p)$ for $p \in [0, 1]$, and we may choose $h$ to be the optimal binary classification error,

$$h((1 - p, p)) = min(p, 1 - p). \tag{4}$$

Another possibility it to use the entropy, $h((1 - p, p)) = p \log p + (1 - p)log(1 - p)$. The average uncertainty of $Y$ is given by $\mathbb{E}_{x \sim X} h(\mathbb{P}(Y|X = x))$. Note that this notion does not depend on a

particular classifier, but reflects the performance the *best* classifier can possibly achieve (under appropriate cost).

The goal of fair representation learning is then to find representations $Z$ that for a given $\gamma \geq 0$ satisfy the constraint (3), and under that constraint minimize the objective $E = E_\theta$ given by

$$E_\theta = \mathbb{E}_{z \sim Z} h(\mathbb{P}(Y|Z = z)). \tag{5}$$

That is, the representation should minimise the optimal $Y$ prediction error (using $Z$) under the fairness constraint.

The curve that associates to every $0 \leq \gamma \leq 1$ the minimum of (5) over all representations $Z$ which satisfy (3) with $\gamma$ is referred to as the *Pareto Front* of the Fairness-Performance trade-off.

In supplementary material Section A we show that for any representation, $\mathbb{E}_{z \sim Z} h(\mathbb{P}(Y|Z = z)) \geq \mathbb{E}_{x \sim X} h(\mathbb{P}(Y|X = x))$, i.e. representations generally decrease or maintain the performance.

### 3.2 Classification and Factorization

In this section we show that the Pareto front of binary classifiers, with accuracy performance and statistical parity fairness measure, can be computed from the Pareto front of representations with total variation fairness measure. In fact, Lemma 3.1 below states that both Pareto fronts amount to the same curve. As discussed in Section 1, this equivalence implies that MIFPO can be used to evaluate fair classifiers, in addition to fair representations.

For a binary classifier $\hat{Y}$ of $Y$, with $(X, A)$ as features. The prediction error is defined as usual by $\epsilon(\hat{Y}) := \mathbb{P}\left(\hat{Y} \neq Y\right)$. The *statistical parity* of $\hat{Y}$ is defined as

$$D_{SP}(\hat{Y}) := \left|\mathbb{P}\left(\hat{Y} = 1|A = 1\right) - \mathbb{P}\left(\hat{Y} = 1|A = 0\right)\right|. \tag{6}$$

**Lemma 3.1.** *Let $\hat{Y}$ be a classifier of $Y$, let the representation uncertainty measure be given by* (4). *Then there is a representation given by a random variable $Z$ on a set $\mathcal{Z}$ with $|\mathcal{Z}| = 2$, such that*

$$\mathbb{E}_{z \sim Z} h(\mathbb{P}(Y|Z = z)) \leq \epsilon(\hat{Y}) \text{ and } \|\mu_0 - \mu_1\|_{TV} \leq D_{SP}(\hat{Y}). \tag{7}$$

*Conversely, for any given representation $Z$, there is a classifier $\hat{Y}$ of $Y$ as a function of $Z$ (and thus of $(X, A)$), such that*

$$\epsilon(\hat{Y}) \leq \mathbb{E}_{z \sim Z} h(\mathbb{P}(Y|Z = z)) \text{ and } D_{SP}(\hat{Y}) \leq \|\mu_0 - \mu_1\|_{TV}. \tag{8}$$

The Proof of Lemma 3.1 is presented in Supplementary Material Section I.

We now describe the Factorization result. Let $f^* : \mathcal{X} \times \mathcal{A} \to \Delta_{\mathcal{Y}}$ be the Bayes optimal classifier of $Y$ given $X, A$. That is, for every $x \in \mathcal{X}, a \in \mathcal{A}$, $f^*(x, a)$ is the conditional distribution of $Y$ given $x, a$, i.e. $f^*(x, a) = \mathbb{P}(Y = \cdot|X = x, A = a)$. Denote by $(X', A)$ a new pair of random variables, taking values in $\Delta_{\mathcal{Y}} \times A$, given by $(X', A) = (f^*(X, A), A)$.

**Lemma 3.2** (Factorization). *For any representation $Z$ of $(X, A)$, there is a representation $Z'$ of $(X', A)$, such that*

$$\mathbb{E}_{z' \sim Z'} h(\mathbb{P}(Y|Z' = z')) \leq \mathbb{E}_{z \sim Z} h(\mathbb{P}(Y|Z = z)) \text{ and } D_{TV}(Z') \leq D_{TV}(Z). \tag{9}$$

In words, for every representation $Z$, we can find a representation $Z'$ that only accesses $(x, a)$ through the value $f^*(x, a)$, and is at least as good in terms of both fairness and performance. Equivalently, this means that any two points $(x_1, a)$ and $(x_2, a)$ with coinciding conditional $Y$ distribution may be treated as identical for the purposes of constructing optimal representations. As a result, to find optimal tradeoffs, we can only consider the representations $Z'$ on the small space $\Delta_{\mathcal{Y}} \times \mathcal{A}$, rather than $Z$ on the much bigger space $\mathcal{X} \times \mathcal{A}$.

Observations related to Lemma 3.2 were made in the context of classification in Kim et al. (2020),Xian et al. (2023), and Wang et al. (2023), which also start from the Bayes optimal classifier. Lemma 3.2 generalizes these observations to representations and to general losses. The proof may be found in Supplementary K.

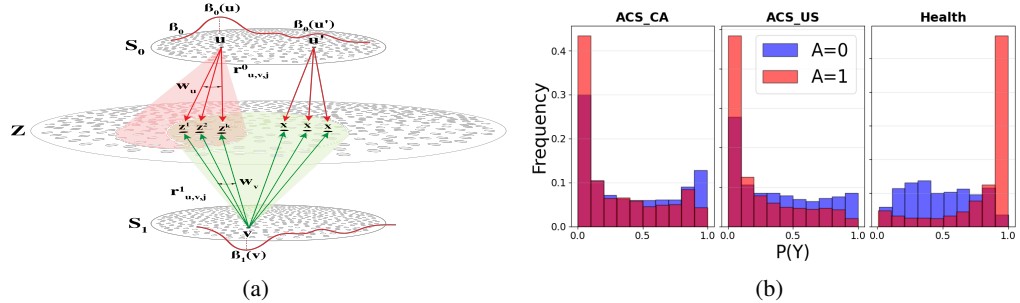

(a)                                                    (b)

Figure 2: (a) The MIFPO Setting (b) Distribution of $P(Y = 1|X, A)$ for each group across datasets.

### 3.3 The Invertibility Theorem

In this section we define the notion of invertibility for representations, and show that considering invertible representations is sufficient for computing the Pareto front.

A representation $Z$ on a set $\mathcal{Z}$ is *invertible* if for every $z \in \mathcal{Z}$ and every $a \in \{0, 1\}$, there is at most one $x \in \mathcal{X}$ such that $\mathbb{P}(Z = z|X = x, A = a) > 0$. In words, a representation is invertible, if any given $z$ can be produced by at most two original features $(x, a)$, and at most one for each value $a$. For $z \in \mathcal{Z}$, we say that an $(x, a)$ is a *parent* of $z$ if $\mathbb{P}(Z = z|X = x, A = a) > 0$.

**Theorem 3.1.** *Let $Z$ be any representation of $(X, A)$ on a set $\mathcal{Z}$. Then there exists an invertible representation $Z'$ of $(X, A)$, on some set $\mathcal{Z}'$, such that*

$$\mathbb{E}_{z' \sim Z'} h(\mathbb{P}(Y|Z' = z')) \leq \mathbb{E}_{z \sim Z} h(\mathbb{P}(Y|Z = z)) \text{ and } D_{TV}(Z') = D_{TV}(Z). \qquad (10)$$

In words, for every representation, we can find an invertible representation of the same data which satisfies at least as good a fairness constraint, and has at least as good performance as the original. In particular, this implies that when one searches for optimal performance representations, it suffices to only search among the invertible ones.

The proof proceeds by observing that if an atom $z \in \mathcal{Z}$ has more than one parent for a fixed $a$, then one can split this atom into two, with each having less parents. However, the details of this construction are somewhat intricate and the full argument can be found in Section C.

Although in this paper we concentrate on the case of binary sensitive variable, we note that Theorem 3.1 may be extended to multi valued attributes, with a similar argument. In that case, invertibility would mean that every $z \in \mathcal{Z}$ would still have at most two parents, $u, v$, corresponding to different values $a, a'$ of $A$.

## 4 The Model Independent Optimization Problem

In this Section we motivate and introduce the MIFPO optimisation problem, and then discuss the full Pareto front computation procedure starting from the raw data.

### 4.1 MIFPO Definition

For the purposes of this Section, we assume that the $x$ feature space $\mathcal{X}$ is finite. In the next section, Section 4.2, we describe how we obtain such finite spaces by using the factorization result and discretizing $\Delta_{\mathcal{Y}}$. Note, however, that the full original, possibly high dimensional feature space $\mathcal{X}$, is never discretized.

Write $S_0 = \{(x, 0) \mid x \in \mathcal{X}\} = \mathcal{X} \times \{0\}$, and similarly $S_1 = \mathcal{X} \times \{1\}$, for the two halves of the full feature space, $\mathcal{X} \times \mathcal{A} = S_0 \cup S_1$.

**Parameters:** The MIFPO parameters model the data distribution and are as follows: **(a)** the probability distributions $\beta_0 \in \Delta_{S_0}$ and $\beta_1 \in \Delta_{S_1}$, on $S_0$ and $S_1$ respectively, modeling $\mathbb{P}((X, A)|A = 0)$ and $\mathbb{P}((X, A)|A = 1)$ respectively, i.e. the distribution of the data features on each sensitive subgroup. **(b)** The subgroup proportions $\alpha_a = \mathbb{P}(A = a)$, and **(c)** the conditional $Y$ distributions,

$\rho_u, \rho_v \in \Delta_{\mathcal{Y}}$, modeling $\rho_u = \mathbb{P}(Y = \cdot | (X, A) = u)$ when $a = 0$ or $\rho_v = \mathbb{P}(Y = \cdot | (X, A) = v)$ when $a = 1$.

**Representation Space:** Perhaps the first question one can ask when constructing a representation of the data as above is: *How large the representation space should be?* We now answer this question using the theory of Section 3.

Fix an integer $k \geq 2$. The representation space $\mathcal{Z}$ will be a finite set which can be written as

$$\mathcal{Z} = S_0 \times S_1 \times [k], \tag{11}$$

where $[k] := \{1, 2, \ldots, k\}$. That is, every point $z \in \mathcal{Z}$ corresponds to some triplet $(u, v, j)$, with $u \in S_0, v \in S_1, j \in [k]$. To explain this choice, recall that by the Invertibility result, we know that we may consider only invertible representations. In such representations, every point $z \in \mathcal{Z}$ is indexed by a pair of parents $(u, v) \in S_0 \times S_1$, suggesting that we may index the points by $S_0 \times S_1$ to begin with. Next, for a given such pair $(u, v)$, we may ask how many points $z$ should have the same pair $(u, v)$ as their parents? In Supplementary Section D, we show that using $k$ points for every pair, we can obtain uniform approximation over all representations. That is, given a degree of approximation $\varepsilon$, Lemma D.1 provides a bound on $k$ which is sufficient to obtain such approximation. While such a bound would clearly depend on $\varepsilon$, we not that it does note depend on the sizes $|S_0|, |S_1|$. These considerations explain the choice of (11) as the representation space. We have used $k = 5$ in all experiments.

**Variables:** The variables of the problem model the representation itself. They will be denoted by $r_{u,v,j}^a$ for $(u, v, j) \in \mathcal{Z}$ and $a \in \mathcal{A}$, and model the probabilities $r_{u,v,j}^a = \mathbb{P}(Z = (u, v, j) | (X, A) = s)$, where either $a = 0$ and $s = u \in S_0$, or $a = 1$ and $s = v \in S_1$ for some $v \in \mathcal{X}$. That is, for $a = 0$, points $u$ transition to $(u, v, j)$ for some $v \in S_1, j \in [k]$, and similarly for $a = 1$, points $v$ transition to $(u, v, j)$ for some $u \in \mathcal{X}, j \in [k]$. This notation preserves our convention that $(u, v, j) \in Z$ has $u$ and $v$ as its only parents. The situation is illustrated in Figure 2(a).

Note that the variables represent probabilities, and thus satisfy the following constraints:

$$r_{u,v,j}^a \geq 0, \quad \forall (u, v, j) \in \mathcal{Z}, \forall a \in \mathcal{A} \tag{12}$$

$$\sum_{v \in S_1, j \in [k]} r_{u,v,j}^0 = 1 \quad \forall u \in S_0 \text{ and } \sum_{u \in S_0, j \in [k]} r_{u,v,j}^1 = 1 \quad \forall v \in S_1 \tag{13}$$

**Performance Objective and Fairness Constraints:** With these preparations, we are ready to write the performance cost (5) in the new notation:

$$E_r = \sum_{z=(u,v,j)} \left[ \alpha_0 \beta_0(u) r_{u,v,j}^0 + \alpha_1 \beta_1(v) r_{u,v,j}^1 \right] \cdot h \left( \frac{\rho_u \alpha_0 \beta_0(u) r_{u,v,j}^0 + \rho_v \alpha_1 \beta_1(v) r_{u,v,j}^1}{\alpha_0 \beta_0(u) r_{u,v,j}^0 + \alpha_1 \beta_1(v) r_{u,v,j}^1} \right). \tag{14}$$

Indeed, observe that due to the structure of our representations, every $z$ has two parents, and we have $\mathbb{P}(Z = z) = \left( \alpha_0 \beta_0(u) r_{u,v,j}^0 + \alpha_1 \beta_1(v) r_{u,v,j}^1 \right)$. Similarly, $\mathbb{P}(Y|Z = z)$ is computed via

$$\mathbb{P}(Y|z) = \sum_{x,a} \mathbb{P}(Y|x, a, z) \mathbb{P}(a, x|z) = \sum_{x,a} \mathbb{P}(Y|x, a) \mathbb{P}(z|a, x) \mathbb{P}(a, x) / \mathbb{P}(z),$$

and substituted inside $h$ to obtain (14). As we show in Supplementary E, the cost (14) is a *concave* function of the variables $r$.

We now proceed to discuss the fairness constraint. Recall that we define $\mu_a(z) = \mathbb{P}(Z = z | A = a)$, for $a \in \{0, 1\}$. For $z = (u, v, j)$ we have then $\mu_a((u, v, j)) = \beta_a(u) r_{u,v,j}^a$, for $a \in \{0, 1\}$, and we can write

$$D_{TV}(Z) = \|\mu_0 - \mu_1\|_{TV} = \frac{1}{2} \sum_z |\mu_0(z) - \mu_1(z)| = \frac{1}{2} \sum_{(u,v,j)} \left| \beta_0(u) r_{u,v,j}^0 - \beta_1(v) r_{u,v,j}^1 \right| \tag{15}$$

and the Fairness constraint, for a given $\gamma \in [0, 1]$, is thus simply

$$\frac{1}{2} \sum_{(u,v,j)} \left| \beta_0(u) r_{u,v,j}^0 - \beta_1(v) r_{u,v,j}^1 \right| \leq \gamma. \tag{16}$$

We now summarise the full MIFPO problem.

**Definition 4.1** (MIFPO). *For a fixed finite ground set $\mathcal{X} \times \mathcal{A} = S_0 \cup S_1$, the problem parameters are the weight $\alpha_0$, the distributions $\beta_0, \beta_1$, on $S_0$ and $S_1$ respectively, and the distributions $\rho_x \in \Delta_{\mathcal{Y}}$ for every $x \in S_0 \cup S_1$. The problem variables are $\left\{ r_{u,v,j}^0, r_{u,v,j}^1 \right\}_{(u,v,j) \in \mathcal{Z}}$ as defined above. We are interested in minimizing the concave function* (14), *subject to the constraints* (12), (13), *and* (16).

The relationship between MIFPO and the Optimal Transport problem is detailed in Supplementary B.

Finally, observe that the MIFPO constraints above *linear*, with (16) being equivalent to two linear inequality constraints. We note that these constraints may be replace by equivalent linear *equality* constraints, via appropriate slack variables, which is more convenient in practice. See Supplementary F for details.

## 4.2  The Full Algorithm

In this Section we summarize the full Pareto front computation algorithm, including the estimation of the MIFPO parameters $\alpha, \beta$ and $\rho$ as discussed above.

Let $D = \{((x_i, a_i), y_i)\}_{i \leq N}$ be the dataset, and write $D_a = \{((x_i, a_i), y_i) \in D \mid a_i = a\}$, so that $D = D_0 \cup D_1$.

The algorithm proceeds in the following steps: **Step 1:** we learn the probability estimators $c_0, c_1 :$ $\mathbb{R}^d \to \Delta_{\mathcal{Y}}$, separately on $D_0$ and $D_1$. These estimators should approximate the optimal Bayes classifier (Section 3.2). Note that such estimation of probabilities is well studied, and is known as *calibration*, see Niculescu-Mizil and Caruana (2005), Kumar et al. (2019), (Berta et al., 2024).

**Step 2:** (Discretization and Parameter estimation) For a given integer $L > 0$, the space $\Delta_{\mathcal{Y}}$ is discretized into $L$ bins. This corresponds to taking the ground sets $S_0, S_1$ in MIFPO to be of size $L$. The data $D_a$ is then mapped into the $L$ bins using $c_a$. The distribution $\beta_a(w)$ is then simply measures the proportion of points $\{c_a(x)\}_{(x,a) \in D_a}$ that fall into bin $w \leq L$. Finally, for every bin $w$, we choose an arbitrary point inside that bin as the representative distribution, $\rho_w$. The parameters $\alpha_a$ are estimated simply by $\alpha_a = |D_a| / |D|$. Note that for binary $Y$, $\Delta_{\mathcal{Y}}$ is simply the interval $[0, 1]$, which is trivial to discretize. See Figure 2(b) for an example of such histograms on real data. We note that one could easily consider more complex discretization schemes, such as clustering, which could be applied efficiently to multi label problems. See Supplementary J for a discussion.

**Step 3:** For a given a fairness threshold $\gamma > 0$, we can now construct the MIFPO instance, Definition 4.1, with $|S_0| = |S_1| = L$, the additional approximation parameter $k$, and $\alpha, \beta, \rho$ as discussed above. As discussed in Section 4.1, we found it sufficient to use $k = 5$ throughout the paper. The MIFPO is then solved using the existing methods, as detailed in Section 5. The full algorithm is schematically show as Algorithm 1, Supplementary H.1.

## 5  Experiments

Our approach requires two main computational components: building calibrated classifiers to evaluate $c_a$, and solving the discrete optimization problem described in Section 4.2. For the calibrated classifier, we have used XGBoost (Chen et al., 2015), with Isotonic Regression calibration, as implemented in sklearn, Pedregosa et al. (2011). Next, as discussed in Sections 1, 4.1, MIFPO is a concave minimisation problem, under linear constraints. To solve its, we have used the DCCP framework and the associated solver (Shen et al., 2016, 2024), which are based on the combination of convex-concave programming (CCP) Lipp and Boyd (2016) and disciplined convex programming, Grant et al. (2006). We note that although local minima are theoretically possible, the above framework is well-established, and the concave structure can be exploited to allow finding optimal solutions in most practical cases, (Shen et al., 2016). In particular, our results do not indicate local minima issues. However, it may also be worth noting that MIFPO could in principle be also solved with the classical branch-and-bound methods, Benson (1995), which may be slower but do guarantee the global optimum solution.

Throughout the experiments, we use the missclassification error loss $h$ given by (4). Additional implementation details may be found in Supplementary Section H.

Our experimental validation of MIFPO encompasses three standard fairness benchmarks: the Health dataset alongside two variants of ACSIncome—one restricted to California (ACSIncome-CA) and

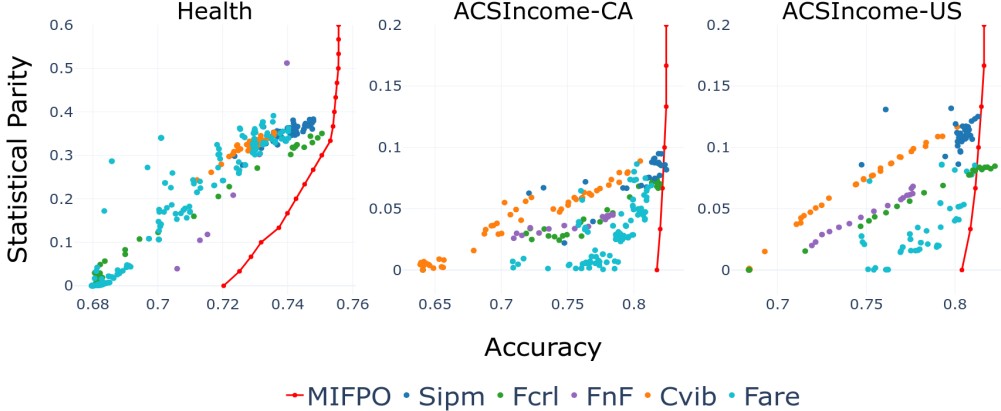

Figure 3: Comparison of fairness-accuracy trade-offs across three benchmark datasets: Health (left), ACSIncome-CA (middle), and ACSIncome-US (right). MIFPO's Pareto front is represented as a solid line with markers. The horizontal axis represents the fairness constraint (statistical parity distance), while the vertical axis shows prediction accuracy.

another spanning the entire United States (ACSIncome-US). In Figure 3 we evaluate MIFPO against five state-of-the-art fair representation techniques: CVIB (Moyer et al., 2019), FCRL (Gupta et al., 2021), FNF (Balunović et al., 2022b), sIPM (Kim et al., 2023), and Fare (Jovanović et al., 2023). Each competitive approach was tuned across diverse hyperparameter settings to generate a spectrum of representations balancing fairness and accuracy. Moreover, we evaluated MIFPO against several fair-classification methods on multiple datasets as presented in the Supplementary Section H.

The empirical results presented in Figure 3 demonstrate MIFPO's effectiveness relative to prior approaches. MIFPO consistently achieves performance equal to or superior than the baseline methods across almost all operating points. Furthermore, MIFPO provides a significant methodological advantage through its ability to characterize the complete Pareto frontier. In the figure, MIFPO's performance is visualized as a solid line with points that trace the entire Pareto front, while competing algorithms are represented as individual points corresponding to different hyperparameter configurations.

## 5.1 Implementation

All evaluations can be found at `https://github.com/bp6725/Efficient-Fair-Pareto-Paper`. The MIFPO algorithm's source code is available in the `https://github.com/bp6725/FairPareto` repository. The algorithm is also implemented as the "FairPareto" Python package on PyPI, which provides a scikit-learn compatible API for computing optimal fairness-performance Pareto fronts. The package supports two usage modes: a tabular mode with automatic classifier training and calibration given a sensitive attribute column, and a second mode for any data type (images, text) where users provide pre-trained classifiers for each sensitive group. This enables researchers to benchmark their fair classification methods against theoretical optimality with minimal code and make informed decisions about fairness-performance trade-offs. The package is open-source, available on PyPI, and includes comprehensive documentation with examples for both tabular and image data.

## 6   Conclusions, Limitations, And Future Work

In this paper we have introduced new fundamental properties of optimal fair representations. In particular, these are the first theoretical results that allow approximation of the Pareto front for arbitrary concave performance measures. We have used these results to develop a model independent procedure for the computation of Fairness-Performance Pareto front from data, demonstrated the procedure on real datasets, and have shown that it may be used as a benchmark for other representation learning algorithms.

We now discuss limitations and a few possible directions for future work. This work primarily concentrated on binary sensitive attribute $A$ and binary $Y$, with the aim to develop the underlying new principles in the simplest case first. As discussed earlier (Sections 1, J), the multi-label case may be treated by more elaborate discretizations. We also noted that the Invertibility Theorem holds for multi valued sensitive attributes as well, which allows to extend the approximation analysis to that case too. Both of these steps, however, would increase the MIFPO problem size. On the other hand, it is also worth noting that this size does not depend directly neither on the feature dimension $d$, nor on the sample size $N$ and thus the problem scales well in that sense.

In view of these observations, we believe it would be of interest to study the following question on the true complexity of the tradeoff evaluation: Suppose we are given access to the Bayes optimal classifier of the data, $f^*$. This encapsulates, in a sense, most of the "continuous" information of the problem. Then, how scalable can Pareto estimation methods be made theoretically, in terms of $|\mathcal{Y}|, |\mathcal{A}|$, while still maintaining controllable approximation bounds?

## Acknowledgments and Disclosure of Funding

This work has received funding from the European Union's Horizon Europe research and innovation programme under grant agreement No. 101070568.

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

# Supplementary Material

## A   Monotonicity of Loss Under Representations

As discussed in Section 3.1, we observe that representations can not increase the performance of the classifier (i.e decrease the loss).

**Lemma A.1.** *For every $(Y, X, A)$, every representation $Z$ as above, and concave $h$,*

$$\mathbb{E}_{z \sim Z} h(\mathbb{P}\left(Y | Z = z\right)) \geq \mathbb{E}_{(x,a) \sim (X,A)} h(\mathbb{P}\left(Y | X = x, A = a\right)). \tag{17}$$

Note that the right hand-side above can be considered a "trivial" representation, $Z = (X, A)$.

In what follows, to simplify the notation we use expressions of the form $\mathbb{P}\left(x, a | z\right)$ to denote the formal expressions $\mathbb{P}\left(X = x, A = a | Z = z\right)$, whenever the precise interpretation is clear from context.

*Proof.* For every value $y \in \mathcal{Y}$, we have

$$\mathbb{P}\left(Y = y | Z = z\right) = \sum_a \int dx \ \mathbb{P}\left(Y = y | x, a, z\right) \frac{\partial \mathbb{P}\left(x | a, z\right)}{\partial x} \mathbb{P}\left(a | z\right) \tag{18}$$

$$= \sum_a \int dx \ \mathbb{P}\left(Y = y | x, a\right) \frac{\partial \mathbb{P}\left(x | a, z\right)}{\partial x} \mathbb{P}\left(a | z\right) \tag{19}$$

$$= \mathbb{E}_{(x,a) \sim (X,A) | Z = z} \mathbb{P}\left(Y = y | x, a\right). \tag{20}$$

Here, on line (18), $\frac{\partial \mathbb{P}(x|a,z)}{\partial x}$ is the density of $\mathbb{P}\left(x | a, z\right)$ with respect to $dx$. Crucially, the transition from (18) to (19) is using the property (1). The transition (19) to (20) is a change of notation. Using

(20) and the concavity of $h$, we obtain

$$\mathbb{E}_{z \sim Z} h(\mathbb{P}(Y|Z = z)) \geq \mathbb{E}_{z \sim Z} \mathbb{E}_{(x,a) \sim (X,A)|Z=z} h(\mathbb{P}(Y|x,a)) \tag{21}$$

$$= \mathbb{E}_{(x,a) \sim (X,A)} h(\mathbb{P}(Y|X = x, A = a)). \tag{22}$$

$\square$

## B  MIFPO and Optimal Transport

In this Section we discuss the relation between the MIFPO minimisation problem, Definition 4.1, and the problem of Optimal Transport (OT). General background on OT may be found in Peyré et al. (2019). We discuss the similarity between OT and the minimisation of (14) under the constraint (16) with $\gamma = 0$, i.e. the perfectly fair case. In this case, (16) is equivalent to the condition $\beta_0(u)r_{u,v,j} = \beta_1(v)r_{v,u,j}$, for all $(u,v,j) \in \mathcal{Z}$. Next, note that thus in is case the expression for $\mathbb{P}(Y|Z = (u,v,j))$ is

$$\frac{\rho_u \alpha_0 \beta_0(u) r_{u,v,j} + \rho_v \alpha_1 \beta_1(v) r_{v,u,j}}{\alpha_0 \beta_0(u) r_{u,v,j} + \alpha_1 \beta_1(v) r_{v,u,j}} = \frac{\rho_u \alpha_0 + \rho_v \alpha_1}{\alpha_0 + \alpha_1}, \tag{23}$$

and *this is independent* of the variables $r$! Therefore we can write the cost (14) as

$$\sum_{u,v} (\alpha_0 + \alpha_1) h\left(\frac{\rho_u \alpha_0 + \rho_v \alpha_1}{\alpha_0 + \alpha_1}\right) \frac{1}{2} \left[\sum_{j \leq k} \beta_0(u) r_{u,v,j} + \beta_1(v) r_{v,u,j}\right]. \tag{24}$$

Note further that for fixed $u, v$, the different $j$'s in this expression play similar roles and could be effectively merged as a single point.

The cost (24) has several similarities with OT. First, in both problems we have two sides, $S_0$ and $S_1$, and we have a certain fixed loss associated with "matching" $u$ and $v$. In case of (24), this loss is $(\alpha_0 + \alpha_1) h\left(\frac{\rho_u \alpha_0 + \rho_v \alpha_1}{\alpha_0 + \alpha_1}\right)$, which describes the information loss incurred by colliding $u$ and $v$ in the representation. And second, similarly to OT, (24) it is *linear* in the variables $r$. Linear programs are conceptually considerably simpler than minimisation of the concave objective (14).

## C  Proof Of Theorem 3.1

In this Section we prove Theorem 3.1.

To keep the notation and the main argument concise, we prove the result under the assumption that $(X, A)$ is finitely supported. Since no assumptions are made on the cardinalities of the supports, the general measurable case follows by standard approximation arguments.

We now introduce the additional notation necessary for the proof. Let $S_0, S_1$ be finite disjoint sets, where $S_a$ represents the values of $(X, A)$ when $A = a$, for $a \in \{0, 1\}$. Denote $S = S_0 \cup S_1$. We are assuming that there is a probability distribution $\zeta$ on $S$, and $A$ is the random variable $A = \mathbb{1}_{\{s \in S_1\}}$. $X$ is defined as taking the values $s \in S$, with $\mathbb{P}(X = s) = \zeta(s)$. Further, the variable $Y$ is defined to take values in a finite set $\mathcal{Y}$, and for every $s \in S$, its conditional distribution is given by $\rho_s \in \Delta_{\mathcal{Y}}$. That is, $\mathbb{P}(Y = y|X = s, A = a) = \rho_s(y) = \rho_{s,a}(y)$. [2] This completes the description of the data model.

For $a \in \{0, 1\}$ we denote $\alpha_a = \mathbb{P}(A = a) = \zeta(S_a)$, and $\beta_a(s) = \mathbb{P}(X = s|A = a)$. Observe that $\beta_a(s) = 0$ if $s \notin S_a$, and $\beta_a(s) = \zeta(s)/\zeta(S_a)$ if $s \in S_a$.

We now describe the representation. The representation will take values in a finite set $\mathcal{Z}$. For every $s \in S$ and $z \in \mathcal{Z}$, let $T_a(z, s) = \mathbb{P}(Z = z|X = s, A = a)$ be the conditional probability of representing $s$ as $z$. $T_a$ are sometimes referred as the *transition kernels* of the representation. For

---

[2]Note that there is a slight redundancy in the notation $\mathbb{P}(Y = y|X = s, A = a)$ here, since $a$ is determined by $s$. However, to retain compatibility with the standard notation, literature, we specify them both. This is similar to the continuous situation, in which although $A$ is technically part of the features, $X$ and $A$ are specified separately.

fixed $(X, A)$, the $T_a$'s fully define the distribution of the representation $Z$ and we shall refer to the representation as $T$ or as $Z$ in interchangeably. Finally, for $a \in \{0, 1\}$ denote

$$\mu_a(z) = T_a \beta_a = \mathbb{P}\left(Z = z | A = a\right) = \sum_{s \in S_a} \beta_a(s) T_a(z, s). \tag{25}$$

With the new notation, a representation $T$ is *invertible* if for every $z \in \mathcal{Z}$ and every $a \in \{0, 1\}$, there is at most one $s \in S_a$ such that $T_a(z, s) > 0$. In words, a representation is invertible, if any given $z$ can be produced by at most two original features $s$, and at most one in each of $S_0$ and $S_1$.

Given a representation $T$ and $z \in \mathcal{Z}$, we say that an $s \in S$ is a *parent* of $z$ if $T_a(z, s) > 0$ for the appropriate $a$.

*Proof Of Theorem 3.1.* Assume $T$ is not invertible. Then there is a $z \in \mathcal{Z}$ which has at least two parents in either $S_0$ or $S_1$. Assume without loss of generality that $z$ has two parents in $S_0$. Let

$$U = \left\{s \in S_0 \ | \ T_0(z, s) > 0\right\}, \quad V = \left\{s \in S_1 \ | \ T_1(z, s) > 0\right\} \tag{26}$$

be the sets of parents of $z$ in $S_0$ and $S_1$ respectively. Chose a point $x \in U$, and denote by $U^r = U \backslash \{x\}$ the remainder of $U$. By assumption we have $|U^r| \geq 1$. We also assume that $|V| > 0$. The easier case $|V| = 0$ will be discussed later.

Now, we construct a new representation, $T'$. The range of $T'$ will be $\mathcal{Z}' = \mathcal{Z} \setminus z \cup \{z', z''\}$. That is, we remove $z$ and add two new points. Denote

$$\kappa = \mathbb{P}\left(x | z, a = 0\right) = \frac{\beta_0(x) T_0(z, x)}{\sum_{s \in U} \beta_0(s) T_0(z, s)}. \tag{27}$$

Then $T'$ is defined as follows:

$$\begin{cases} T'_a(h, s) = T_a(h, s) & \text{for all } a \in \{0, 1\}, \text{ all } s \in S \text{ and all } h \in \mathcal{Z} \setminus \{z\} \\ T'_0(z', x) = T_0(z, x) \\ T'_0(z'', u) = T_0(z, u) & \text{for all } u \in U^r \\ T'_1(z', v) = \kappa T_1(z, v) & \text{for all } v \in V \\ T'_1(z'', v) = (1 - \kappa) T_1(z, v) & \text{for all } v \in V. \end{cases} \tag{28}$$

All values of $T'$ that were not explicitly defined in (28) are set to 0. In words, on the side of $S_0$, we move all the parents of $z$ except $x$ to be the parents of $z''$, while $z'$ will have a single parent, $x$. On the $S_1$ side, both $z'$ and $z''$ will have the same parents as $z$, with transitions multiplied by $\kappa$ and $1 - \kappa$ respectively. The multiplication by $\kappa$ is crucial for showing both inequalities in (10).

Note that $T'$ can be though of as splitting $z$ into $z'$ and $z''$, such that $z'$ has one parent on the $S_0$ side, and $z''$ has strictly less parents than $z$ had. Once we show that $T'$ satisfies (10), it is clear that by induction we can continue splitting $T'$ until we arrive at an invertible representation which can no longer be split, thus proving the Lemma.

In order to show (10) for $T'$, we will sequentially show the following claims:

$$\mathbb{P}\left(z\right) = \mathbb{P}\left(z'\right) + \mathbb{P}\left(z''\right) \text{ and } \mathbb{P}\left(z'\right) = \kappa \mathbb{P}\left(z\right) \tag{29}$$

$$\mathbb{P}\left(a | z\right) = \mathbb{P}\left(a | z'\right) = \mathbb{P}\left(a | z''\right) \text{ for } a \in \{0, 1\} \tag{30}$$

$$\begin{cases} \text{for } a = 1 & \mathbb{P}\left(s | z, a\right) = \mathbb{P}\left(s | z', a\right) = \mathbb{P}\left(s | z'', a\right) \quad \forall s \in S \\ \text{for } a = 0, s \in U^r & \begin{cases} \mathbb{P}\left(x | z', a\right) = 1 & \mathbb{P}\left(x | z'', a\right) = 0 \\ \mathbb{P}\left(s | z', a\right) = 0 & \mathbb{P}\left(s | z'', a\right) = (1 - \kappa)^{-1} \mathbb{P}\left(s | z, a\right) \end{cases} \end{cases} \tag{31}$$

$$\mathbb{P}\left(Y | z\right) = \kappa \mathbb{P}\left(Y | z'\right) + (1 - \kappa) \mathbb{P}\left(Y | z''\right) \tag{32}$$

$$\mathbb{P}\left(z\right) h(\mathbb{P}\left(Y | z\right)) \geq \mathbb{P}\left(z'\right) h(\mathbb{P}\left(Y | z'\right)) + \mathbb{P}\left(z''\right) h(\mathbb{P}\left(Y | z''\right)) \tag{33}$$

$$|\mu_0(z) - \mu_1(z)| = |\mu'_0(z') - \mu'_1(z')| + |\mu'_0(z'') - \mu'_1(z'')|. \tag{34}$$

Here the probabilities involving $z', z''$ refer to the representation $T'$. Observe that the left hand side of (33) is the contribution of $z$ to the performance cost $\mathbb{E}_{t \sim Z} h(\mathbb{P}\left(Y | Z = t\right))$ of $T$, while the right

hand side of (33) is the contribution of $z'$, $z''$ to the performance cost of $T'$. Since all other elements $t \in \mathcal{Z}$ have identical contributions, this shows the first inequality in (10). Similarly, recall that

$$\|\mu_0 - \mu_1\|_{TV} = \frac{1}{2} \sum_{t \in \mathcal{Z}} |\mu_0(t) - \mu_1(t)|, \tag{35}$$

and thus the left hand side of (34) is the contribution of $z$ to $\|\mu_0 - \mu_1\|_{TV}$, with the right hand side being the contribution of $z'$, $z''$ to $\|\mu_0' - \mu_1'\|_{TV}$, therefore yielding the claim $\|\mu_0' - \mu_1'\|_{TV} = \|\mu_0 - \mu_1\|_{TV}$.

**Claim** (29): By definition,

$$\mathbb{P}(z') = \alpha_0 \beta_0(x) T_0'(z', x) + \alpha_1 \sum_{s \in V} \beta_1(s) T_1'(z', s) \tag{36}$$

$$= \alpha_0 \beta_0(x) T_0(z, x) + \kappa \alpha_1 \sum_{s \in V} \beta_1(s) T_1(z, s) \tag{37}$$

$$= \kappa \alpha_0 \sum_{s \in U} \beta_0(s) T_0(z, s) + \kappa \alpha_1 \sum_{s \in V} \beta_1(s) T_1(z, s) \tag{38}$$

$$= \kappa \mathbb{P}(z). \tag{39}$$

Similarly, by definition we have

$$\mathbb{P}(z'') = \alpha_0 \sum_{s \in U^r} \beta_0(s) T_0(z, s) + (1 - \kappa) \kappa \alpha_1 \sum_{s \in V} \beta_1(s) T_1(z, s), \tag{40}$$

and summing this with (37), we obtain $\mathbb{P}(z) = \mathbb{P}(z') + \mathbb{P}(z'')$.

**Claim** (30): Note that it is sufficient to prove the claim for $a = 0$ since the probabilities sum to 1. Write

$$\mathbb{P}(a = 0 | z) = \frac{\mathbb{P}(a = 0, z)}{\mathbb{P}(z)} \tag{41}$$

$$= \frac{\alpha_0 \sum_{s \in U} \beta_0(s) T_0(z, s)}{\mathbb{P}(z)} \tag{42}$$

$$= \frac{\kappa \cdot \alpha_0 \sum_{s \in U} \beta_0(s) T_0(z, s)}{\kappa \cdot \mathbb{P}(z)} \tag{43}$$

$$= \frac{\alpha_0 \beta_0(x) T_0(z, x)}{\mathbb{P}(z')} \tag{44}$$

$$= \mathbb{P}(a = 0 | z'). \tag{45}$$

Similarly,

$$\mathbb{P}(a = 0 | z) = \frac{\mathbb{P}(a = 0, z)}{\mathbb{P}(z)} \tag{46}$$

$$= \frac{\alpha_0 \sum_{s \in U} \beta_0(s) T_0(z, s)}{\mathbb{P}(z)} \tag{47}$$

$$= \frac{(1 - \kappa) \cdot \alpha_0 \sum_{s \in U} \beta_0(s) T_0(z, s)}{(1 - \kappa) \cdot \mathbb{P}(z)} \tag{48}$$

$$= \frac{\alpha_0 \sum_{s \in U^r} \beta_0(s) T_0(z, s)}{\mathbb{P}(z'')} \tag{49}$$

$$= \mathbb{P}(a = 0 | z''). \tag{50}$$

**Claim** (31): For $a = 1$, let us show $\mathbb{P}\left(s|z, a\right) = \mathbb{P}\left(s|z', a\right)$.

$$\mathbb{P}\left(s|z, a = 1\right) = \frac{\alpha_1 \beta_1(s) T_1(z, s)}{\alpha_1 \sum_{s' \in V} \beta_1(s') T_1(z, s')} \tag{51}$$

$$= \frac{\kappa \alpha_1 \beta_1(s) T_1(z, s)}{\kappa \alpha_1 \sum_{s' \in V} \beta_1(s') T_1(z, s')} \tag{52}$$

$$= \frac{\alpha_1 \beta_1(s) T_1'(z, s)}{\alpha_1 \sum_{s' \in V} \beta_1(s') T_1'(z, s')} \tag{53}$$

$$= \mathbb{P}\left(s|z', a = 1\right). \tag{54}$$

The statement $\mathbb{P}\left(s|z, a\right) = \mathbb{P}\left(s|z'', a\right)$ is shown similarly. Next, for $a = 0$, we have $\mathbb{P}\left(x|z', a\right) = 1$ and $\mathbb{P}\left(x|z'', a\right) = 0$ by the definition of the coupling $T'$. Moreover, for $s \in U^r$, $\mathbb{P}\left(s|z', a\right) = 0$ also follows by the definition of $T'$. Finally, write

$$\mathbb{P}\left(s|z'', a = 0\right) = \frac{\alpha_0 \beta_0(s) T_0'(z'', s)}{\sum_{s \in U^r} \alpha_0 \beta_0(s') T_0'(z'', s')} \tag{55}$$

$$= \frac{\alpha_0 \beta_0(s) T_0(z, s)}{\sum_{s \in U^r} \alpha_0 \beta_0(s') T_0(z, s')} \tag{56}$$

$$= \frac{\alpha_0 \beta_0(s) T_0(z, s)}{(1 - \kappa) \sum_{s \in U} \alpha_0 \beta_0(s') T_0(z, s')} \tag{57}$$

$$= (1 - \kappa)^{-1} \mathbb{P}\left(s|z, a = 0\right). \tag{58}$$

**Claim** (32): We first observe that for any representation (and any z),

$$\mathbb{P}\left(Y = y|Z = z\right) = \frac{\sum_{s, a} \mathbb{P}\left(Y = y, s, a, z\right)}{\mathbb{P}\left(z\right)} \tag{59}$$

$$= \frac{\sum_{s, a} \mathbb{P}\left(Y = y|s, a\right) \mathbb{P}\left(s, a, z\right)}{\mathbb{P}\left(z\right)} \tag{60}$$

$$= \sum_a \mathbb{P}\left(a|z\right) \left[\sum_{s \in S_a} \mathbb{P}\left(Y = y|s, a\right) \mathbb{P}\left(s|a, z\right)\right], \tag{61}$$

where we have used the property (1) for the transition (59)-(60). Now, using (31), for $a = 1$ we have

$$\sum_{s \in S_1} \mathbb{P}\left(Y = y|s, a = 1\right) \mathbb{P}\left(s|a = 1, z\right) = \sum_{s \in S_1} \mathbb{P}\left(Y = y|s, a = 1\right) \mathbb{P}\left(s|a = 1, z'\right)$$

$$= \sum_{s \in S_1} \mathbb{P}\left(Y = y|s, a = 1\right) \mathbb{P}\left(s|a = 1, z''\right). \tag{62}$$

For $a = 0$, we have for $z'$ using (31):

$$\sum_{s \in S_0} \mathbb{P}\left(Y = y|s, a = 1\right) \mathbb{P}\left(s|a = 1, z'\right) = \mathbb{P}\left(Y = y|x, a = 1\right). \tag{63}$$

For $a = 0$ and $z''$ we have

$$\sum_{s \in S_0} \mathbb{P}\left(Y = y|s, a = 1\right) \mathbb{P}\left(s|a = 1, z''\right) = \sum_{s \in U^r} \mathbb{P}\left(Y = y|s, a = 1\right) \mathbb{P}\left(s|a = 1, z''\right) \tag{64}$$

$$= (1 - \kappa)^{-1} \sum_{s \in U^r} \mathbb{P}\left(Y = y|s, a = 1\right) \mathbb{P}\left(s|a = 1, z\right) \tag{65}$$

where we have used (31) again on the last line.

Combining (62),(63),(65), and using (30) and the general expression (61), we obtain the claim (32).

**Claim** (33): This follows immediately from (32) by using (29) and the concavity of $h$.

**Claim (34):** By definition, for every representation, $\mu_a(z) = \mathbb{P}\left(Z = z|a\right) = \frac{\mathbb{P}(z|a)\mathbb{P}(z)}{\mathbb{P}(a)}$. Thus, using (29),(30) we have for $a \in \{0, 1\}$,

$$\mu_a(z') = \kappa\mu_a(z) \text{ and } \mu_a(z'') = (1 - \kappa)\mu_a(z), \tag{66}$$

which in turn yields (34).

It remains only to recall that we have derived (33),(34) under the assumption that $|V| > 0$. That is, we assumed that the point $z$ which fails invertability on $S_0$ has some parents in $S_1$. The case when $|V| = 0$, i.e. there are no parents in $S_1$ can be treated using a similar argument, but is much simpler. Indeed, in this case one can simply split $z$ into $z'$ and $z''$ and splitting the $S_0$ weight between them as before, without the need to carefully balance the interaction of probabilities with $S_1$ via $\kappa$.

$\square$

## D  Uniform Approximation and Two Point Representations

As discussed in Sections 1,4.1, we are interested in showing that all optimal invertible representations, no matter which, and no matter on which set $\mathcal{Z}'$, can be approximated using a representation with the following property: For every $u \in S_0, v \in S_1$, there are at most $k$ points $z \in \mathcal{Z}$ that have $(u, v)$ as parents, see Figure 2(a). Here $k$ would depend only on the desired approximation degree, but not on $\mathcal{Z}'$, or on the exact representation we are approximating. We therefore refer to this result as the Uniform Approximation result. Its implications for practical use were discussed in Section 4.1.

The notation used in this Section was introduced in the beginning of Section C.

To proceed with the analysis, in what follows we introduce the notion of two-point representation. The main result is given as Lemma D.1 below.

This Section uses the notation of Section 3.3. Let $T$ be an invertible representation, let $u \in S_0, v \in S_1$ be some points, and denote by $\mathcal{Z}_{uv} = \left\{z^j\right\}_1^k$ the set of all points $z \in \mathcal{Z}$ which have $u$ and $v$ as parents. Denote by

$$w_u = \sum_{j=1}^{k} \beta_0(u)T_0(z^j, u) \text{ and } w_v = \sum_{j=1}^{k} \beta_1(v)T_1(z^j, v) \tag{67}$$

the total weights of $\beta_0$ and $\beta_1$ transferred by the representation from $u$ and $v$ respectively to $\mathcal{Z}_{uv}$. Recall that $\rho_u, \rho_v$ denote the distributions of $Y$ conditioned on $u, v$. We call the situation above, i.e. the collection of numbers $\left(\left\{\beta_0(u)T_0(z^j, u)\right\}_{j\leq k}, \left\{\beta_1(v)T_1(Z^j, v)\right\}_{j\leq k}\right)$, a *two point representation*, since it describes how the weight from the points $u, v$ is distributed in the representation, independently of the rest of the representation. The contribution of $\mathcal{Z}_{uv}$ to the global performance cost is

$$E_{uv,T} := \sum_{j\leq k} \mathbb{P}\left(z^j\right) h(\mathbb{P}\left(Y|z^j\right)) \tag{68}$$

$$= \sum_{j\leq k} \left(\alpha_0\beta_0(u)T_0(z^j, u) + \alpha_1\beta_1(v)T_1(z^j, v)\right) h\left(\frac{\alpha_0\beta_0(u)T_0(z^j, u)\rho_u + \alpha_1\beta_1(v)T_1(z^j, v)\rho_v}{\alpha_0\beta_0(u)T_0(z^j, u) + \alpha_1\beta_1(v)T_1(z^j, v)}\right), \tag{69}$$

while its contribution to the fairness condition is

$$F_{uv,T} = \frac{1}{2}\sum_{j\leq k} \left|\beta_0(u)T_1(z^j, u) - \beta_1(v)T_1(z^j, v)\right|. \tag{70}$$

Let us now consider two extreme cases of two-point representations. Assume that the total amounts of weight to be represented, $w_u, w_v$ are fixed. The first case is when $k = 1$, and this is the maximum fairness case, since in this case the weights $w_u, w_v$ overlap as much as possible. Indeed, the contributions to the fairness penalty and performance cost in this case are

$$|w_u - w_v| \text{ and } (\alpha_0 w_u + \alpha_1 w_v) h\left(\frac{\alpha_0 w_u \rho_u + \alpha_1 w_v \rho_v}{\alpha_0 w_u + \alpha_1 w_v}\right) \tag{71}$$

respectively. The other extreme case is when $w_u$ and $w_v$ do not overlap at all. This case be realised with $k = 2$, by sending all $w_u$ to $z^1$ and all $w_v$ to $z^2$. The fairness and performance contributions would be

$$w_u + w_v \text{ and } \alpha_0 w_u \cdot h(\rho_u) + \alpha_1 w_v \cdot h(\rho_v), \tag{72}$$

respectively. Note that the fairness penalty is the maximum possible, while the performance cost is the minimum possible (indeed, this is the cost before the representation, and any representation can only increase it, by Lemma A.1). We thus observed that each two points $u, v$, with fixed total weight $w_u, w_v$, can have their own Pareto front of performance-fairness. One could, in principle, fix a threshold $\gamma_{uv}, |w_u - w_v| \leq \gamma_{uv} \leq w_u + w_v$ for the fairness penalty (70), and obtain a performance cost between that in (71) and (72). However, it is not clear how large the number of points $k$ should be in order to realise such intermediate representations. In the following Lemma we show that one can uniformly approximate all the points on the two-point Pareto front using a fixed number of points, that depends only on the function $h$. This means that in practice one can choose a certain number $n$ of $z$ points, and have guaranteed bounds on the possible amount of loss incurred with respect to all representations of all other sizes.

**Lemma D.1.** *For every $\varepsilon > 0$, there a number $n = n_\varepsilon$ depending only on the function $h$, with the following property: For every two-point representation $\left( \{\beta_0(u) T_0(z^j, u)\}_{j \leq k}, \{\beta_1(v) T_1(Z^j, v)\}_{j \leq k} \right)$, with total weights $w_u, w_v$, there is a two point representation $T'$ on a set $\mathcal{Z}'_{u,v}$, with the same total weights, such that $|\mathcal{Z}'_{uv}| \leq n$, and such that*

$$F_{uv,T'} \leq F_{uv,T} \text{ and } E_{uv,T'} \leq E_{uv,T} + 2(w_u + w_v)\varepsilon. \tag{73}$$

*Proof.* To aid with brevity of notation, define for $j \leq k$

$$c_0^j = \alpha_0 \beta_0(u) T_0(z^j, u), \quad c_1^j = \alpha_1 \beta_1(v) T_1(z^j, v). \tag{74}$$

Then we can write

$$E_{uv,T} = \sum_{j \leq k} (c_0^j + c_1^j) \cdot h\left(\frac{c_0^j \rho_u + c_1^j \rho_v}{c_0^j + c_1^j}\right), \quad F_{uv,T} = \frac{1}{2} \sum_{j \leq k} \left|\Lambda c^j\right|, \tag{75}$$

where $\Lambda$ is the vector $\Lambda = (\alpha_0^{-1}, -\alpha_1^{-1})$, $c^j = (c_0^j, c_1^j)$, and $\Lambda c^j$ is the inner product of the two.

Observe that the cost $E_{uv,T}$ depends on $c^j$ mainly through the fractions $\frac{c_0^j}{c_0^j + c_1^j}$. Our strategy thus would be to approximate all $k$ of such fractions by a $\delta$-net of a size independent of $k$. To this end, set

$$p^j = \frac{c_0^j}{c_0^j + c_1^j} \tag{76}$$

and define $h_{uv} : [0,1] \to \mathbb{R}$ by

$$h_{uv}(p) = h(p\rho_u + (1-p)\rho_v). \tag{77}$$

Since $h$ is continuous (by assumption), and defined on a compact set, it is *uniformly* continuous, and so is $h_{uv}$. By definition, this means there is a $\delta > 0$ such that for all $p, p'$ with $|p - p'| \leq \delta$, it holds that $|h_{uv}(p) - h_{uv}(p')| \leq \varepsilon$. Let us now choose $\{x_i\}_{i=1}^n$ to be a $\delta$ net on $[0,1]$. For every $i \leq n$ set

$$\Gamma_i = \left\{ j \mid |p^j - x_i| \leq \delta, \text{ and } i \text{ is minimal with this property} \right\}. \tag{78}$$

That is, $\Gamma_i$ is the set of indices $j$ such that $p^j$ is approximated by $x^i$. Using $x_i$ we construct the representation $T'$ as follows: For $a \in \{0, 1\}$ set

$$c_a'^i = \sum_{j \in \Gamma_i} c_a^j. \tag{79}$$

For $n$ new points, $z^i \in \mathcal{Z}'_{uv}$, set $T'_0(z'^i, u) = c_0'^i / \beta_0(u)$, $T'_1(z'^i, v) = c_1'^i / \beta_1(v)$. Note that the total weights are preserved, $\sum_{i \leq n} c_0'^i = w_u$ and $\sum_{i \leq n} c_1'^i = w_v$.

Next, for every $j \in \Gamma_i$ we have

$$\left| \frac{c_0^j}{c_0^j + c_1^j} - x_i \right| \leq \delta. \tag{80}$$

Thus

$$\left| \frac{c_0'^i}{c_0'^i + c_1'^i} - x_i \right| = \left| \frac{\sum_{j \in \Gamma_j} \left[ c_0^j - (c_0^j + c_1^j) x_i \right]}{c_0'^i + c_1'^i} \right| \leq \frac{\sum_{j \in \Gamma_j} \delta(c_0^j + c_1^j)}{c_0'^i + c_1'^i} = \delta. \tag{81}$$

Next, observe that by the construction of $x_i$,

$$\left| E_{uv,T} - \sum_i (c_0'^i + c_1'^i) h_{uv}(x_i) \right| = \left| \sum_i \sum_{j \in \Gamma_i} (c_0^j + c_1^j) h_{uv}(p^j) - \sum_i (c_0'^i + c_1'^i) h_{uv}(x_i) \right| \tag{82}$$

$$\leq \sum_i \sum_{j \in \Gamma_i} (c_0^j + c_1^j) \varepsilon \tag{83}$$

$$= (w_u + w_v) \varepsilon. \tag{84}$$

In addition,

$$\left| E_{uv,T'} - \sum_i (c_0'^i + c_1'^i) h_{uv}(x_i) \right| = \left| \sum_i (c_0'^i + c_1'^i) h_{uv}\left( \frac{c_0'^i}{c_0'^i + c_1'^i} \right) - \sum_i (c_0'^i + c_1'^i) h_{uv}(x_i) \right| \tag{85}$$

$$\leq (w_u + w_v) \varepsilon, \tag{86}$$

where we have used (81) in the last transition.

Combining the two inequalities yields the second part of (73),

$$|E_{uv,T'} - E_{uv,T}| \leq 2(w_u + w_v) \varepsilon. \tag{87}$$

Finally, note that

$$\sum_i \left| \Lambda c'^i \right| = \sum_i \left| \sum_{j \in \Gamma_j} \Lambda c^j \right| \tag{88}$$

$$\leq \sum_i \sum_{j \in \Gamma_j} \left| \Lambda c^j \right| \tag{89}$$

$$= \sum_j \left| \Lambda c^j \right|, \tag{90}$$

yielding the first part of, and thus completing the proof of, statement (73).

It remains to observe that above we have used a $\delta$ net for $h_{uv}$, which depends on $\rho_u, \rho_v$. However, we can directly build an appropriate $\delta$-net in full range of $h$, the simplex $\Delta_{\mathcal{Y}}$, which would produce bounds valid for all $u, v$. Indeed, let $\delta'$ be such that $|h(\nu) - h(\nu)| \leq \varepsilon$ for all $\mu, \nu \in \Delta_{\mathcal{Y}}$ with $\|u - v\|_1 \leq \delta'$. Observe that the map $p \mapsto p\rho_v + (1-p)\rho_u$ is 2-Lipschitz from $\mathbb{R}$ to $\Delta_{\mathcal{Y}}$ equipped with the $\|\cdot\|_1$ norm, for any $u, v \in \Delta_{\mathcal{Y}}$. Thus, choosing $\delta = \frac{1}{2}\delta'$, we have $|h_{uv}(p) - h_{uv}(p')| \leq \varepsilon$ if $|p - p'| \leq \delta$. This completes the proof of the Lemma. □

# E  Concavity Of $E_r$

Note that the variables $r$ appear in (14) both as coefficients multiplying $h$ and inside the arguments of $h$, in a fairly involved manner. Nevertheless, the cost turns out to still retain an interesting structure, as it is *concave*, if $h$ is. We record this in the following Lemma.

**Lemma E.1.** *If $h : \Delta_{\mathcal{Y}} \to \mathbb{R}$ is concave, then of every $\rho_1, \rho_2 \in \Delta_{\mathcal{Y}}$ the function $g : \mathbb{R}^2 \to \mathbb{R}$, given by $g((c_1, c_2)) = (c_1 + c_2)h(\frac{c_1\rho_1 + c_2\rho_2}{c_1 + c_2})$ is concave.*

*Proof.* It is sufficient to show that for every $c, c' \in \mathbb{R}^2$, we have $g((c + c')/2) \geq \frac{1}{2}(g(c) + g(c'))$. To this end, define the map $F : \mathbb{R}^2 \to \Delta_{\mathcal{Y}}$ by

$$F(c) = \frac{c_1\rho_1 + c_2\rho_2}{c_1 + c_2} \tag{91}$$

and note that

$$F((c+c')/2) = F(c)\frac{c_1 + c_2}{c_1 + c_2 + c_1' + c_2'} + F(c')\frac{c_1' + c_2'}{c_1 + c_2 + c_1' + c_2'}. \tag{92}$$

It then follows that

$$g((c+c')/2) = \frac{1}{2}(c_1 + c_2 + c_1' + c_2')h(F((c+c')/2)) \tag{93}$$

$$= \frac{1}{2}(c_1 + c_2 + c_1' + c_2')h\left(F(c)\frac{c_1 + c_2}{c_1 + c_2 + c_1' + c_2'} + F(c')\frac{c_1' + c_2'}{c_1 + c_2 + c_1' + c_2'}\right) \tag{94}$$

$$\geq \frac{1}{2}(c_1 + c_2 + c_1' + c_2')\left[\frac{c_1 + c_2}{c_1 + c_2 + c_1' + c_2'}h(F(c)) + \frac{c_1' + c_2'}{c_1 + c_2 + c_1' + c_2'}h(F(c'))\right] \tag{95}$$

$$= \frac{1}{2}\left(g(c) + g(c')\right). \tag{96}$$

$\square$

## F  MIFPO Equality Constraint

As noted in the main text, although the inequality constraint (16) is convex in the variables $r$, and can be incorporated directly into most optimisation frameworks, it may be significantly more convenient to work with *equality* constraints. Using the following Lemma, we can find equivalent equality constraints in a particularly simple form.

**Lemma F.1.** *Let* $\mu_0, \mu_1 \in \Delta_\mathcal{Z}$ *be two probability distributions over* $\mathcal{Z}$ *and fix some* $\gamma \geq 0$. *If* $\|\mu_0 - \mu_1\|_{TV} = \gamma$ *then there exist* $\phi_0, \phi_1 \in \Delta_\mathcal{Z}$ *such that* $\mu_0 + \gamma\phi_0 = \mu_1 + \gamma\phi_1$. *In the other direction, if there exist* $\phi_0, \phi_1 \in \Delta_\mathcal{Z}$ *such that* $\mu_0 + \gamma\phi_0 = \mu_1 + \gamma\phi_1$, *then* $\|\mu_0 - \mu_1\|_{TV} \leq \gamma$.

The proof may be found in Section G.

As consequence of this result, if we find distributions $\phi_0, \phi_1 \in \Delta_\mathcal{Z}$ such that $\mu_0 + \gamma\phi_0 = \mu_1 + \gamma\phi_1$ holds, then we know that (16) also holds, and conversely, if (16) holds, then distributions as above exist.

Using this observation, we introduce new variables, $\phi^0_{u,v,j}$ and $\phi^1_{u,v,j}$, for every $(u, v, j) \in \mathcal{Z}$, which correspond to $\phi_0((u, v, j))$ and $\phi_1((u, v, j))$ respectively. These variables will be required to satisfy the following constraints:

$$\phi^0_{u,v,j} \geq 0, \quad \phi^1_{u,v,j} \geq 0 \quad \forall (u, v, j) \in \mathcal{Z} \tag{97}$$

$$\sum_{u,v,j} \phi^0_{u,v,j} = 1 \text{ and } \sum_{u,v,j} \phi^1_{u,v,j} = 1 \tag{98}$$

$$\beta_0(u)r^0_{u,v,j} + \gamma\phi^0_{u,v,j} = \beta_1(v)r^0_{v,u,j} + \gamma\phi^1_{u,v,j} \quad \forall (u, v, j) \in \mathcal{Z}. \tag{99}$$

Here the first two lines encode the fact that $\phi_0, \phi_1$ are probabilities, while the third line encodes the fairness constraint, as discussed above.

## G  Additional Proofs

Proof Of Lemma F.1

*Proof.* For this proof it is more convenient to work with the $\ell_1$ norm $\|\cdot\|_1$ directly. Recall that $\|\mu_0 - \mu_1\|_{TV} = \frac{1}{2}\|\mu_0 - \mu_1\|_1$ and that

$$\|\mu_0 - \mu_1\|_1 = \sum_z |\mu_0(z) - \mu_1(z)|. \tag{100}$$

Assume that $\|\mu_0 - \mu_1\|_1 = 2\gamma$. Define the functions $\bar{\phi}_0(z) = \mathbb{1}_{\{\mu_1 \geq \mu_0\}}(z) \cdot (\mu_1(z) - \mu_0(z))$ and $\bar{\phi}_1(z) = \mathbb{1}_{\{\mu_0 \geq \mu_1\}}(z) \cdot (\mu_0(z) - \mu_1(z))$. Note that we then have

$$\sum_z \bar{\phi}_0(z) = \sum_z \bar{\phi}_1(z) = \gamma. \tag{101}$$

Indeed, define

$$\eta(z) = \mathbb{1}_{\{\mu_0 \geq \mu_1\}}(z) \cdot \mu_1(z) + \mathbb{1}_{\{\mu_1 \geq \mu_0\}}(z) \cdot \mu_0(z). \tag{102}$$

Clearly, $\eta + \bar{\phi}_0 = \mu_1$ and thus $\sum_z \eta(z) + \bar{\phi}_0(z) = 1$. Similarly, $\sum_z \eta(z) + \bar{\phi}_1(z) = 1$. Therefore we have

$$\sum_z \bar{\phi}_0(z) = \sum_z \bar{\phi}_1(z). \tag{103}$$

Note also that we can write

$$2\gamma = \|\mu_0 - \mu_1\|_1 = \sum_z |\mu_0(z) - \mu_1(z)| = \sum_z \left( \bar{\phi}_0(z) + \bar{\phi}_1(z) \right), \tag{104}$$

which combined with (103) yields (101).

Next, we can also directly verify that

$$\mu_0 + \bar{\phi}_0 = \mu_1 + \bar{\phi}_1, \tag{105}$$

and thus setting $\phi_0 = \gamma^{-1}\bar{\phi}_0, \phi_1 = \gamma^{-1}\bar{\phi}_1$ completes the proof of this direction.

In the other direction, given $\phi_0, \phi_1 \in \Delta_{\mathcal{Z}}$ such that $\mu_0 + \gamma\phi_0 = \mu_1 + \gamma\phi_1$, we have

$$\sum_z |\mu_0(z) - \mu_1(z)| = \gamma \sum_z |\phi_1(z) - \phi_0(z)| \leq 2\gamma, \tag{106}$$

thus completing the proof. □

# H  Experiments

This section describes additional evaluation details and experiments with fair classifiers. In SectionH.1 we provide the main algorithm figure, and discuss technical implementation details. Section H.2 contains the comparison to a number of fair classifiers, and Section H.3 discusses implementation of the entropy cost $h$ within the DCCP framework.

## H.1  Implementations and computational details

---
**Algorithm 1** MIFPO Implementation

---
**Input:** Data $\{(x_i, a_i, y_i)\}_{i \leq N}$, integers $L, k$.
  For $a \in \{0, 1\}$ denote $X_a = \{x_i \mid a_i = a\}$.
**1.** Learn *calibrated* classifiers
  $c_0, c_1 : \mathbb{R}^d \to [0, 1]$, such that
  $c_a(x) \sim \mathbb{P}(Y = 1 | X = x, A = a)$
**2.** Construct the histograms $\{\beta_a(l)\}_{l=1}^L$,
  $a \in \{0, 1\}$, for the sets
  $H_a = \{c_a(x) \mid (x, a) \in D_a\} \subset [0, 1]$.
  Choose bin representatives $\{\rho_l\}_{l=1}^L$
**3.** Solve MIFPO, given by Definition 4.1,
  with parameters $k$ and
  $\{\beta_a(l)\}_{l=1}^L, \{\rho_l\}_{l=1}^L, \alpha_a = |\{i \mid a_i = a\}| / N$.

---

The Pareto front evaluation requires two main parts - building a calibrated classifier required for evaluating $c_a = P(Y|X, A = a)$, and later solving the optimization problem MIFPO (see Algorithm 1).

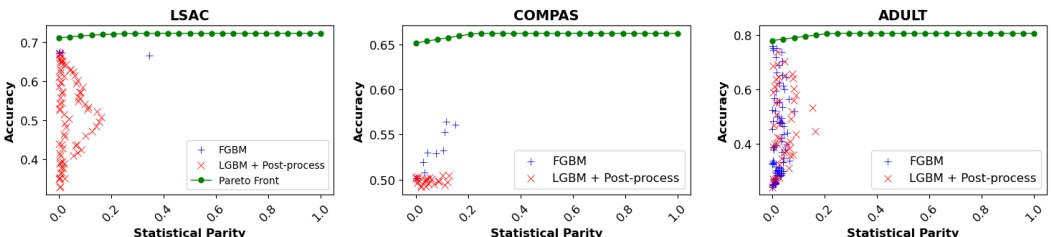

Figure 4: Comparing common fair classification pipelines to the MIFPO Pareto front, for LSAC, COMPAS, and ADULT datasets. For FGBM and LGBM+Post Process methods, each point represents a trade-off obtained at a single hyper-parameter configuration.

For a calibrated classifier, we are using standard model calibration. Model calibration is a well-studied problem where we fit a monotonic function to the probabilities of some base model so that the probabilities will reflect real probabilities, that is, $P(Y|X)$. Here, we used Isotonic regression (Berta et al., 2024) for model calibration with XGBoost (Chen et al., 2015) as the base model. For training the XGBoost model, a GridSearchCV approach is employed to find the best hyperparameters from a specified parameter grid, using 3-fold cross-validation.

The experiments were conducted on a system with an Intel Core i9-12900KS CPU (16 cores, 24 threads), 64 GB of RAM, and an NVIDIA GeForce RTX 3090 GPU.

## H.2 Additional Evaluations

Following the equivalence between the Pareto fronts of fair binary classification and representations for the accuracy cost (Section 3.1), we evaluated the accuracy-fairness trade-off for some common fair classification algorithms. For our evaluation, we selected the most widely used algorithms based on GitHub repository stars and citation counts in the literature, demonstrating the importance of our proposed method in comparison to common approaches. We also evaluate FairFront, a more recent approach introduced in Wang et al. (2023).

Fair classifiers generally fall into pre-processing, in-processing, and post-processing categories. Pre- and post-processing types often utilize standard classifiers as part of their fair classification pipeline. We specifically evaluated two widely adopted algorithms, representing different categories: FairGBM Cruz et al. (2023): An in-processing method where a boosting trees algorithm (LightGBM) is subjected to pre-defined fairness constraints. Balanced-Group-Threshold Jang et al. (2021): A post-processing method which adjusts the threshold per group to obtain a fairness criterion. For FairGBM, we used the original implementation provided by the authors. For Balanced-Group-Threshold post-processing, we utilized implementations available via Aequitas Saleiro et al. (2018), a popular bias and fairness audit toolkit.

We conducted our evaluation using three of the most common datasets in this field, which are known to have relevance to real-world decision-making processes: the Adult dataset (income prediction), COMPAS (recidivism prediction), and LSAC (law school admission).

It is important to note that, as a rule, common fairness classification methods are not designed to control the fairness-accuracy trade-off explicitly. Instead, in most cases, these methods rely on rerunning the algorithm for a wide range of hyperparameter settings, in the hope that different hyperparameters would result in different fairness-accuracy trade-off points. However, there typically is no direct known and controlled relation between hyperpatameters and the obtained fairness-accuracy trade-off. For FairGBM, we utilized the hyperparameter ranges specified in the original paper, Cruz et al. (2023). In the case of the balancing post-processing method, we conducted a grid search over the full range of all possible hyperparameters to ensure a comprehensive analysis.

Figure 4 shows the MIFPO computed Pareto front, and all hyperparameter runs of the two algorithms above, with accuracy evaluated on the test set. These experiments demonstrate the following two points: **(a)** The standard classifiers achieve a considerably lower accuracy than what is theoretically possible at a given fairness level. **(b)** the existing methods are also unable to present solutions for

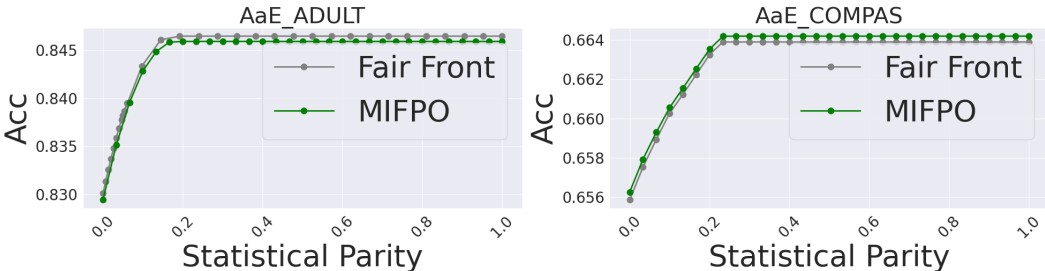

Figure 5: Comparing FairFront to MIFPO accuracy-fairness tradeoff on two curated datasets.

the full range of the statistical parity values. The values from the FGBM and the post-processing algorithms all have statistical parity $\leq 0.2$. Similarly to to the case of fair representations, these results emphasize the limitations of current fair classifiers in achieving optimal trade-offs between accuracy and fairness across the full range of fairness values.

Additionally, we add a comparison to FairFront (Wang et al., 2024), which, similarly to our work, depends on estimates of $P(Y|X)$. See the discussion in Section 2. We note, however, that the utility of this comparison is limited due to the very strict setup defined in that paper, which cannot be applied to natural datasets. Specifically: 1) The implementation in Wang et al. (2023) requires creating a finite discrete space *by binning over the full $\mathbb{R}^d$ feature space*[3], which allows for perfect modeling of the distribution $P(Y|X)$ (by simple counting). This is not normally possible on real datasets in practice. 2) The binning itself was performed manually for each dataset separately, and we were unable to discern the logic behind the selected parameters. 3) To allow reasonable binnig, the true dimensions of the data are manually reduced, again by picking features manually for each dataset. Due to these issues, it was practically impossible to apply the method in Wang et al. (2023) to other standard datasets, or even to the datasets used in Wang et al. (2023) but with standard features.

Nonetheless, for the sake of comparison, we compared MIFPO and FairFront on the same restricted, preprocessed version of the datasets that FairFront used, with the same bin-based probability estimator. The results are shown in Figure 5.

### H.3 Minimization of Concave Functions under Convex Constraints and the Entropy Loss

As described in figure 5, we used the disciplined convex concave programming (DCCP) framework and the associated solver, (Shen et al., 2016, 2024) for solving the concave minimization with convex constraints problem.

Minimizing concave functions under convex constraints is a common problem in optimization theory. Unlike convex optimization where global minima can be readily found, in concave minimization problems we only know that the local minimas lie on the boundaries of the feasible region defined by the convex constraints. While techniques such as branch-and-bound algorithms, cutting plane methods, and heuristic approaches are often employed, here we used the framework of DCCP which gain a lot of popularity in recent years.

The DCCP framework extends disciplined convex programming (DCP) to handle nonconvex problems with objective and constraint functions composed of convex and concave terms. The idea behind a "disciplined" methodology for convex optimization is to impose a set of conventions inspired by basic principles of convex analysis and the practices of convex optimization experts. These "disciplined" conventions, while simple and teachable, allow much of the manipulation and transformation required for analyzing and solving convex programs to be automated. DCCP builds upon this idea, providing an organized heuristic approach for solving a broader class of nonconvex problems by combining DCP principles with convex-concave programming (CCP) methods, and is implemented as an extension to the CVXPY package in Python.

While convenient, the use of the disciplined framework bears some limitations. Mainly, generic operations like element-wise multiplication are not under the allowed set of operations (and for

---

[3]This is completely different from the discretizaton of $\Delta_{\mathcal{Y}}$ used in MIFPO. The space $\mathbb{R}^d$ is the feature space, and is much larger than $\Delta_{\mathcal{Y}}$.

obvious reasons), which limits the usability. Notice, that for the prediction accuracy measure $h(p) = \min(p, 1 - p)$ this is not a problem, but for the entropy classification error $h((1 - p, p)) = -p \log p - (1 - p) \log(1 - p)$, this is more challenging. Nevertheless, here we show that the standard DCCP framework allows for entropy classification error.

**Lemma H.1.** *Let $a = (1 - \alpha)\mathcal{V}(v)r_{v,z}$ and $b = \alpha \cdot \mathcal{U}(u)r_{u,z}$, with $p_v = p_a$ and $p_u = p_b$.*

*We can write the cost function under the entropy accuracy error as:*

$$(a + b) \cdot \text{entropy} \left( \frac{a \cdot p_a + b \cdot p_b}{a + b} \right)$$
$$= \text{entropy}(a \cdot (1 - p_a) + b \cdot (1 - p_b)) + \text{entropy}(a \cdot p_a + b \cdot p_b) - \text{entropy}(a + b)$$

*Proof.*

$$\text{entropy}(x) = \text{entropy}(x, 1 - x) = x \left[ \log(x) - \log(1 - x) \right] + \log(1 - x)$$

Thus,

$$\text{entropy} \left( \frac{a \cdot p_a + b \cdot p_b}{a + b} \right) = \frac{a \cdot p_a + b \cdot p_b}{a + b} \left[ \log \left( \frac{a \cdot p_a + b \cdot p_b}{a + b} \right) - \log \left( 1 - \frac{a \cdot p_a + b \cdot p_b}{a + b} \right) \right]$$
$$+ \log \left( 1 - \frac{a \cdot p_a + b \cdot p_b}{a + b} \right)$$
$$= \frac{a \cdot p_a + b \cdot p_b}{a + b} \left[ \log(a \cdot p_a + b \cdot p_b) - \log(a + b - a \cdot p_a - b \cdot p_b) \right]$$
$$+ \log(a + b - a \cdot p_a - b \cdot p_b) - \log(a + b)$$

Hence :

$$(a + b) \cdot \text{entropy} \left( \frac{a \cdot p_a + b \cdot p_b}{a + b} \right) = (a + b) \cdot \left[ \frac{a \cdot p_a + b \cdot p_b}{a + b} \left[ \log(a \cdot p_a + b \cdot p_b) - \log(a + b - a \cdot p_a - b \cdot p_b) \right] \right]$$
$$+ (a + b) \cdot \log(a + b - a \cdot p_a - b \cdot p_b)$$
$$- (a + b) \cdot \log(a + b)$$

$$= \text{entropy}(a \cdot p_a + b \cdot p_b) - (a \cdot p_a + b \cdot p_b) \cdot \log(a + b - a \cdot p_a - b \cdot p_b)$$
$$+ (a + b) \cdot \log(a + b - a \cdot p_a - b \cdot p_b) - \text{entropy}(a + b)$$

Finally, the expression can be written as:

$$= \text{entropy}(a + b - a \cdot p_a - b \cdot p_b) + \text{entropy}(a \cdot p_a + b \cdot p_b) - \text{entropy}(a + b)$$

$$= \text{entropy}(a \cdot (1 - p_a) + b \cdot (1 - p_b)) + \text{entropy}(a \cdot p_a + b \cdot p_b) - \text{entropy}(a + b)$$

□

Given the element-wise entropy function $x \cdot (1 - x)$ is with known characteristics and under the dccp framework, we can use the entropy error for our cost using :

$$= \text{entropy}((1 - \alpha)\mathcal{V}(v)r_{v,z} \cdot (1 - p_v) + \alpha \cdot \mathcal{U}(u)r_{u,z} \cdot (1 - p_u))$$
$$- \text{entropy}((1 - \alpha)\mathcal{V}(v)r_{v,z} \cdot p_v + \alpha \cdot \mathcal{U}(u)r_{u,z} \cdot p_u)$$
$$- \text{entropy}((1 - \alpha)\mathcal{V}(v)r_{v,z} + \alpha \cdot \mathcal{U}(u)r_{u,z})$$

# I Fair Classifiers As Fair Representations

As discussed in Section 3.1, Pareto front of binary classifiers with statistical parity can be computed from the Pareto front of representations with total variation fairness distance. In this Section we provide the proof of this result, Lemma 3.1. The Lemma and the related notation are restated here for convenience.

For a binary classifier $\hat{Y}$ of $Y$, its prediction error is defined as usual by $\epsilon(\hat{Y}) := \mathbb{P}\left(\hat{Y} \neq Y\right)$. The statistical parity *distance* of $\hat{Y}$ is defined as

$$\Delta_{SP}(\hat{Y}) := \left| \mathbb{P}\left(\hat{Y} = 1 | A = 1\right) - \mathbb{P}\left(\hat{Y} = 1 | A = 0\right) \right|. \tag{107}$$

Let the uncertainy measure $h$ be defined by (4). Note that the first part of the Lemma does use the special properties of this $h$ and does not necessarily hold for other costs $h$.

**Lemma I.1.** *Let $\hat{Y}$ be a classifier of $Y$. Then there is a representation given by a random variable $Z$ on a set $\mathcal{Z}$ with $|\mathcal{Z}| = 2$, such that*

$$\mathbb{E}_{z \sim Z} h(\mathbb{P}\left(Y | Z = z\right)) \leq \epsilon(\hat{Y}) \text{ and } \|\mu_0 - \mu_1\|_{TV} \leq \Delta_{SP}(\hat{Y}). \tag{108}$$

*Conversely, for any given representation $Z$, there is a classifier $\hat{Y}$ of $Y$ as a function of $Z$ (and thus of $(X, A)$), such that*

$$\epsilon(\hat{Y}) \leq \mathbb{E}_{z \sim Z} h(\mathbb{P}\left(Y | Z = z\right)) \text{ and } \Delta_{SP}(\hat{Y}) \leq \|\mu_0 - \mu_1\|_{TV}. \tag{109}$$

*Proof.* Let us begin with the second part of the Lemma, inequalities (109). Given a representation $Z$, $\epsilon(\hat{Y}) \leq \mathbb{E}_{z \sim Z} h(\mathbb{P}\left(Y | Z = z\right))$ follows since $\mathbb{E}_{z \sim Z} h(\mathbb{P}\left(Y | Z = z\right))$ is the error of the optimal classifier of $Y$ as a function of $Z$. We choose $\hat{Y}$ to be such an optimal classifier and thus satisfy the above inequality, with equality. Next, the second inequality in (109) holds for for *any* classifier $\hat{Y}$ derived from $Z$. The argument below is a slight generalisation of the argument in Madras et al. (2018). Define $f(z) = \mathbb{P}\left(\hat{Y} = 1 | Z = z\right)$. Note that for $a \in \{0, 1\}$, $\mathbb{P}\left(\hat{Y} = 1 | A = a\right) = \int f(z) d\mu_a(z)$. Thus

$$\mathbb{P}\left(\hat{Y} = 1 | A = 0\right) - \mathbb{P}\left(\hat{Y} = 1 | A = 1\right) = \int f(z) d\mu_0(z) - \int f(z) d\mu_1(z) \tag{110}$$

$$\leq \sup_{g | \|g\| \leq 1} \left| \int g(z) d\mu_0(z) - \int g(z) d\mu_1(z) \right| \tag{111}$$

$$= \|\mu_0 - \mu_1\|_{TV}, \tag{112}$$

where we have used $|f| \leq 1$ in the second line. Repeating the argument also for $\mathbb{P}\left(\hat{Y} = 1 | A = 1\right) - \mathbb{P}\left(\hat{Y} = 1 | A = 0\right)$, we obtain the second inequality in (109).

We now turn to the first statement, (108). Let $\hat{Y}$ be a classifier of $Y$ as a function of $(X, A)$. Observe that thus by definition $\mathbb{P}\left(\hat{Y} | X, A\right)$ induces a distribution on the set $\{0, 1\}$, and thus may be considered as a representation $Z := \hat{Y}$ of $(X, A)$ on that set. We now relate the properties of this $Z$ as a representation to the quantities $\epsilon(\hat{Y})$ and $\Delta_{SP}(\hat{Y})$. Similarly to the argument above, the first part of (108) follows since $\mathbb{E}_{z \sim Z} h(\mathbb{P}\left(Y | Z = z\right))$ is the best possible error over all classifiers. For the second part, note that since $\hat{Y}$ is binary, we have

$$\mathbb{P}\left(\hat{Y} = 1 | A = 0\right) - \mathbb{P}\left(\hat{Y} = 1 | A = 1\right) = -\mathbb{P}\left(\hat{Y} = 0 | A = 0\right) + \mathbb{P}\left(\hat{Y} = 0 | A = 1\right). \tag{113}$$

It follows that

$$\|\mu_0 - \mu_1\|_{TV} = \frac{1}{2} \sum_{v \in \{0,1\}} \left| \mathbb{P}\left(\hat{Y} = v | A = 0\right) - \mathbb{P}\left(\hat{Y} = v | A = 1\right) \right| \tag{114}$$

$$= \left| \mathbb{P}\left(\hat{Y} = 1 | A = 0\right) - \mathbb{P}\left(\hat{Y} = 1 | A = 1\right) \right| \tag{115}$$

$$= \Delta_{SP}(\hat{Y}), \tag{116}$$

where we have used (113) for the second to third line transition. This completes the proof of the second part of (108). □

# J   Computational Complexities

In this Section we discuss alternative discretization schemes for the MIFPO algorithm. We also discuss various complexity aspects of the classification algorithms Xian et al. (2023) and Wang et al. (2023) and relate them to the complexity of MIFPO.

In Section 4.2 we described a data independent discretization of $\Delta_{\mathcal{Y}}$ by binning. While effective for small label sets $\mathcal{Y}$, larger sets would require a different approach. One alternative is to cluster the data instead of binning $\Delta_{\mathcal{Y}}$ itself. Indeed, by choosing cluster centers $\{\eta_i\}_1^M \subset \Delta_{\mathcal{Y}}$, such that each data point $\{f^*(x,a)\}_{(x,a)\in D}$ is well approximated by one the centers (or just most points are approximated), we can guarantee arbitrarily good approximation of the true Pareto front. The cardinality $M$ of such clustering would depend on the intrinsic dimension of the data, which we typically expect to be lower than the full dimension of $\Delta_{\mathcal{Y}}$, due to the Manifold Hypothesis, Fefferman et al. (2016).

Another possibility is to not use any explicit discretization, and instead to use each data point as a separate bin (equivalently, we use the points themselves as the cluster centers $\{\eta_i\}$). In this case, the complexity scales with the size of the dataset, but not with the dimensions of $\Delta_{\mathcal{Y}}$. The MIFPO construction in Section 4.1 implies that MIFPO in this case would have $O(N^2)$ variables for a dataset of size $N$. While not applicable for large $N$, this is similar to the complexity of a variety of often used algorithms. Classical example of such complexity is the Spectral Clustering. We also observe that Xian et al. (2023), a recent state of the art fairness classification algorithm mentioned above, also has such complexity.

Indeed, the approach in Xian et al. (2023) involves the computation of a certain transportation plan between data points, and the encoding of such plans also requires $O(N^2)$ variables. Thus problem sizes occuring in MIFPO would be smaller or equal to those in Xian et al. (2023), despite the fact that MIFPO is solving a considerably more general problem (see Section 2).

Finally, the algorithm in Wang et al. (2023) involves optimization in the space of confusion matrices, with dimensions of size $|\mathcal{A}| \cdot |\mathcal{Y}| \times |\mathcal{Y}|$. As discussed in Section 2, the reduction of the problem to confusion matrices of possible due to special properties of the classification problem and the accuracy loss.

The algorithm in Wang et al. (2023), FairFront, is an iterative algorithm, where each iteration involves solving a certain *difference-of-convex* (DC) program which is constructed from a full dataset. The class of DC programs is equivalent to that convex-concave programs considered in this paper (see Shen et al. (2016)). In fact, similarly to MIFPO, the algorithm in Wang et al. (2023) also uses DCCP, Shen et al. (2016), although applied to a different problem.

In each iteration, the solution of DC program is then used to add new constraints to a certain main convex program. While it is proved that asymptotically this process converges to the optimal front, there are no bounds on the number of iterations. This may lead to the convex solver crashing due to too many constraints, and in fact we have observed such crashes in our evaluation.

To summarize [4], MIFPO involves solving *one* convex-concave problem, with size which may be independent of the data size. In contrast, FairFront involves iteratively solving convex-concave problems and a main convex program, where the number of terms in the objective of each convex-concave problem scales with the size of the data, and the number of constraints grows in the convex problem grows with iterations, thus making the iterations progressively harder.

# K   Factorization

In this Section we the factorization result, Lemma 3.2. We restate the result for convenience.

Recall that $f^*(x,a)$ denotes the Bayes optimal classifier of $Y$, i.e. $f^* : \mathcal{X} \times \mathcal{A} \to \Delta_{\mathcal{Y}}$ is given by

$$f^*(x,a) := \mathbb{P}\left(Y = \cdot | X = x, A = a\right). \tag{117}$$

We define a new variable $X'$, with values in $\Delta_{\mathcal{Y}}$, by $X' := f^*(X, A)$.

---

[4]In this Section we have discussed the theoretical complexity aspects of FairFront. Additional details pertaining to the official implementation of Wang et al. (2023) may be found in Section H.2.

**Lemma K.1** (Factorization). *For any representation $Z$ of $(X, A)$, there is a representation $Z'$ of $(X', A)$, such that*

$$\mathbb{E}_{z' \sim Z'} h(\mathbb{P}(Y|Z' = z')) = \mathbb{E}_{z \sim Z} h(\mathbb{P}(Y|Z = z)) \text{ and } D_{TV}(Z') \leq D_{TV}(Z). \qquad (118)$$

*Proof.* The representation $Z'$ of $X', A$ will be defined as follows: For $\sigma \in \Delta_{\mathcal{Y}}, a \in \mathcal{A}, z \in \mathcal{Z}$ set:

$$\mathbb{P}(Z' = z|X' = \sigma, A = a) := \mathbb{P}(Z = z|X' = \sigma, A = a) \qquad (119)$$
$$= \mathbb{P}(Z = z|f^*(X, a) = \sigma, A = a) \qquad (120)$$

where the second line is the definition and is added for clarity. To see intuition behind this definition note that we have

$$\mathbb{P}(Z' = z|X' = \sigma, A = a) = \mathbb{P}(Z = z|f^*(X, a) = \sigma, A = a) \qquad (121)$$
$$= \sum_{x \in \mathcal{X}} \mathbb{P}(Z = z|X = x, f^*(x, a) = \sigma, A = a) \mathbb{P}(X = x|f^*(x, a) = \sigma, A = a)$$
$$= \sum_{x \in \mathcal{X}} \mathbb{P}(Z = z|X = x, A = a) \mathbb{P}(X = x|f^*(x, a) = \sigma, A = a)$$
$$\qquad (122)$$

In words, for a fixed $a$, to compute $\mathbb{P}(Z' = z|X' = \sigma, A = a)$ we effectively collect all $x$ such that $f^*(x, a) = \sigma$ and average all of their representations. Equivalently, all points $x$ with the same $\sigma$ are merged into one point, and their representations summed up according to their relative weight.

We will now show that neither the performance nor the fairness condition change under this operation.

Since $D_{TV}(Z)$ is defined solely in terms of the distributions $\mathbb{P}(Z = \cdot|A = a)$, to show that $D_{TV}(Z) = D_{TV}(Z')$ it is enough to show that

$$\mathbb{P}(Z = \cdot|A = a) = \mathbb{P}(Z' = \cdot|A = a) \quad \forall a \in \mathcal{A}. \qquad (123)$$

To this end, we have

$$\mathbb{P}(Z' = z|A = a) = \sum_{\sigma} \mathbb{P}(Z' = z|X' = \sigma, A = a) \mathbb{P}(X' = \sigma|A = a) \qquad (124)$$
$$= \sum_{\sigma} \mathbb{P}(Z = z|X' = \sigma, A = a) \mathbb{P}(X' = \sigma|A = a) \qquad (125)$$
$$= \mathbb{P}(Z = z|A = a). \qquad (126)$$

where the transition (124) to (125) is by the definition of $Z'$. Thus we have shown that $D_{TV}(Z) = D_{TV}(Z')$.

Next, note that the above argument implies also that $\mathbb{P}(Z = z) = \mathbb{P}(Z' = z)$. Thus, in order to show the performance equality,

$$\mathbb{E}_{z' \sim Z'} h(\mathbb{P}(Y|Z' = z')) = \mathbb{E}_{z \sim Z} h(\mathbb{P}(Y|Z = z)), \qquad (127)$$

it is enough to show that $\mathbb{P}(Y|Z' = z) = \mathbb{P}(Y|Z = z)$ for every $z \in \mathcal{Z}$. Further, again since $\mathbb{P}(Z = \cdot) = \mathbb{P}(Z' = \cdot)$, we can show that

$$\mathbb{P}(Y = y, Z' = z) = \mathbb{P}(Y = y, Z = z) \qquad (128)$$

for all $z \in \mathcal{Z}, y \in \mathcal{Y}$. Write

$$\mathbb{P}\left(Y = y, Z = z\right) \tag{129}$$

$$= \sum_a \sum_x \mathbb{P}\left(Y = y, X = x, A = a, Z = z\right) \tag{130}$$

$$= \sum_a \sum_x \mathbb{P}\left(Y = y | X = x, A = a, Z = z\right) \mathbb{P}\left(X = x, A = a, Z = z\right) \tag{131}$$

$$= \sum_a \sum_x \mathbb{P}\left(Y = y | X = x, A = a, Z = z\right) \mathbb{P}\left(Z = z | X = x, A = a\right) \mathbb{P}\left(X = x, A = a\right) \tag{132}$$

$$= \sum_a \sum_x \mathbb{P}\left(Y = y | X = x, A = a\right) \mathbb{P}\left(Z = z | X = x, A = a\right) \mathbb{P}\left(X = x, A = a\right) \tag{133}$$

$$= \sum_a \sum_\sigma \sum_{x | f^*(x,a) = \sigma} \mathbb{P}\left(Y = y | X = x, A = a\right) \mathbb{P}\left(Z = z | X = x, A = a\right) \mathbb{P}\left(X = x, A = a\right)$$
$$\tag{134}$$

$$= \sum_a \sum_\sigma \sigma(y) \mathbb{P}\left(X' = \sigma, A = a\right) \sum_{x | f^*(x,a) = \sigma} \mathbb{P}\left(Z = z | X = x, A = a\right) \frac{\mathbb{P}\left(X = x, A = a\right)}{\mathbb{P}\left(X' = \sigma, A = a\right)}$$
$$\tag{135}$$

$$= \sum_a \sum_\sigma \sigma(y) \mathbb{P}\left(X' = \sigma, A = a\right) \sum_{x | f^*(x,a) = \sigma} \mathbb{P}\left(Z = z | X = x, A = a\right) \mathbb{P}\left(X = x | X' = \sigma, A = a\right))$$
$$\tag{136}$$

$$= \sum_a \sum_\sigma \sigma(y) \mathbb{P}\left(X' = \sigma, A = a\right) \mathbb{P}\left(Z' = z | X' = \sigma, A = a\right) \tag{137}$$

$$= \sum_a \sum_\sigma \sigma(y) \mathbb{P}\left(Z' = z, X' = \sigma, A = a\right) \tag{138}$$

$$= \sum_a \sum_\sigma \mathbb{P}\left(Y = y | X' = \sigma, A = a\right) \mathbb{P}\left(Z' = z, X' = \sigma, A = a\right) \tag{139}$$

$$= \mathbb{P}\left(Y = y, Z' = z\right). \tag{140}$$

Here, the transition (132)-(133) is due to the independence condition (1). On line (134) we split the sum over $x$ into sum over subsets $\{x \mid f^*(x,a) = \sigma\}$ and an outer some over all $\sigma$. The transition (136)-(137) is due to the equality (122)-(121). Finally, for the transition (137)-(138), we have used the fact that

$$\sigma(y) = \mathbb{P}\left(Y = y | X' = \sigma, A = a\right), \tag{141}$$

which holds by definition of $X'$. Similarly to the earlier discussion on merging of $x$ with similar value of $f^*$, the above argument proceeded by regrouping the summation over $x$ by the value of $f^*(x, a)$. The computation thus showed that this process yields the definition of $Z'$. In particular, this regrouping process and equation (141) explain why the space $\Delta_{\mathcal{Y}}$ is special and all representations may be factored through it. This completes the proof of the Lemma. $\qquad \square$

In the above argument we have used the summation over $\sigma$, i.e. $\sum_\sigma \ldots$. This is formally possible when $(X, A)$ has a discreet distribution. The full general case may be obtained simply by replacing the summation by integration and conditioning on $\sigma$ by the general conditional expectation operator.

