# OpenReview forum: "Efficient Fairness-Performance Pareto Front Computation"
_NeurIPS.cc/2025/Conference — NeurIPS 2025 spotlight_

### Official Review · Reviewer_DJzQ · 2025-06-28

**Clarity:** 3
**Significance:** 3
**Originality:** 4
**Rating:** 4
**Confidence:** 3

**Summary:**

This paper proposed an interesting approach to address the fairness–performance trade-off in representation learning. Based on the structural properties of optimal fair representations, this approach reduces Pareto-front computation to a compact concave minimization problem (MIFPO) over discrete variables, which can be solved via off-the-shelf disciplined convex–concave programming (DCCP). The key insight is that we can factor the high-dimensional representation problem through the Bayes optimal classifier to a small simplex (Factorization Lemma), then further restrict our attention to invertible representations (Invertibility Theorem). Empirically, MIFPO provides competitive fairness–accuracy pareto comparing five other methods (CVIB, FCRL, FNF, sIPM, Fare) on Health and ACSIncome datasets. It provides the full Pareto front rather than isolated operating points .

The main contributions are
1) By proving the Factorization Lemma and the Invertibility Theorem, the paper presents an approach of reducing an intractable infinite-dimensional search over representations to a finite concave minimization problem
2) The paper offers the first model independent algorithm that applies to any concave performance measure including accuracy, log loss, etc.
3) The authors create a model independent procedure that can be used to compute the fairness-performance pareto from data. Experimental results show its effectiveness comparing to other methods.

**Questions:**

1) In Section 4.1, what criteria or heuristics should guide the selection of the number of bins L and the replication factor k? Is it possible to quantify the trade-off between approximation error and solver runtime as a function of L and k?

2) What are the size of the datasets used? Can you provide cpu and memory usage? Will the proposed approach scale as the dataset size increases?

3) The solution of MIFPO relies on the DCCP convex-concave solver, which in general only guarantees convergence to a local minimum. Does solution quality matter for this approach?

**Ethical Concerns:**

["NO or VERY MINOR ethics concerns only"]

**Final Justification:**

The author provided a comprehensive rebuttal.

**Limitations:**

Yes.

**Paper Formatting Concerns:**

No.

**Quality:**

3

**Strengths And Weaknesses:**

The paper has following main strengths:
1) The proposed approach is the first to prove fundamental  theorems that produce exact Pareto front reduction.
2) Unlike prior research restricted to non-differentiable accuracy, the proposed framework accommodates any concave performance measure including log-loss and Brier score by using a unified concave minimization formulation .
Decoupling Representation Learning
3) The proposed approach not only finds a single point but traces the entire fairness–performance frontier, offering deep insight into the trade-off that point-wise methods cannot provide

The main weakness include
1)  The theory and algorithm are developed for binary sensitive attributes and targets and extension to multi-class or group settings is deferred and may increase problem size significantly
2) DCCP handles moderate sizes, but concave minimization can stuck into local minima and also may not scale to large-scale problem without specialized global solvers
3) The reported empirical study is limited to three variants of a socioeconomic task. There is no assessment on other sensitive features like race and gender or on higher-dimensional data like text

---

> ### Author Rebuttal · Authors · 2025-07-31
>
> Thank you for the favourable review!
>
> We were glad to hear that the reviewer found our approach interesting, and that the theoretical contributions, and this work being the first to treat general concave performace losses in particular,  were cited as stregths.
>
>
> We now discuss the questions and other points mentioned in the review.
> Please let us know wether these notes help clarify the questions in the review.
>
>
> >    In Section 4.1, what criteria or heuristics should guide the selection of the number of bins L and the replication factor k? Is it possible to quantify the trade-off between approximation error and solver runtime as a function of L and k?
>
> We have found the empirical results to be  fairly insensitive to the values of L and k, as long as L is high enough.
>
>
> **Values of L**:
> Consider for instance L = 50, with corresponding bin width being 0.02. For such bin width, it is possible to show that the maximal error in accuracy and/or statistical parity would be 0.02, and a similar stability result would hold for general losses too. The general result would state that if two distributions on the simplex are close in the transportation (or earth mover) distance, then the corresponding Pareto fronts are close, with distance depending on the modulus of continuity of the perfromance metric. Because with growing bin count L we can uniformly approximate any distribution on the simplex, this result would imply the above statement for L=50.
>
> Should we add this stability result to the paper?
>
> **Values of k**:
> One consequence of Lemma 3.1 is that when  the accuracy performance metric is used, k=2 is theoretically sufficient (see also the discussion on lines 127-139, and 184-188). Interestingly, however, we have found that using k=5 *slightly* increases the performance, which is likely due to non-convex nature of the optimization.
> We have then used k=5 throughout.
>
> >Is it possible to quantify the trade-off between approximation error and solver runtime as a function of L and k?
>
> Because the approximation is non-convex, we are not aware of any possible approach to theoretically quantify the relation between L and runtime.
>
> Empirically, on the other hand, since the  values of L do not influence the performance strongly above a certain level, and since the runtimes were not a bottleneck, we have not fully investigated this  relationship.
> However, here we provide some results about the order of magnitude of the time complexity results for a single point on the Pareto front for different k and l, avraged on few datasets:
>
> | l\k | 2 | 5 |
> |-----|---|---|
> | **10** | 0.18 sec | 0.41 sec |
> | **15** | 0.24 sec | 0.53 sec |
> | **20** | 0.24 sec  | 0.55 sec |
> | **30** | 0.37 sec | 0.82 sec |
> | **50** | 1.06 sec | 4.66 sec |
>
>
> As mentioned above, we used $k=5$ and $L=50$ throughout, thus generating a full Pareto front (30 points) would be under 3  minutes.
>
>
>
> >  What are the size of the datasets used? Can you provide cpu and memory usage? Will the proposed approach scale as the dataset size increases?
>
> The dataset sizes are as follows:
> | | #features | #samples |
> |-----|---|---|
> |Health | 52 | $1.22 \cdot 10^5$  |
> |ACSIncome-US | 11 | $10^6$ |
> |ACSIncome-CA | 11  | $1.15 \cdot 10^5$ |
>
>
> As we detailed above, the runtimes for each of these datasets were all under 3 minutes.
>
> **Scaling**:
> Although due to space constraints we have only discussed this briefly in the paper (lines 357-359), we believe the sclaing  with the dataset size is one of the strongest points of our approach.
>
> Indeed, note that the MIFPO problem itself, as defined in Section 4, does not depend **at all** neither on the dataset size $N$ nor on the data features dimension $d$. The only way in which MIFPO dependes on the data is through the distributions on the simplexes, provided by the probability estimators, as discribed in Step 2 in Section 4.2.
> Because we discretize those distributions, there would be dependence on $L$, but not on $N$!
>
> That said, there are two ways in which $N$ and $d$ may still influence the results.
> **(a)** First, generally, as the dimension grows, it may be harder to construct the probability estimator. Depending on the model used, the dataset size may be either an issue or an asset here. We note however, that calibration/probablility estimation, is the easier part. The harder part is contructiong a base classifier, and that problem is general, i.e., not specifically a fairness problem.
> **(b)** If we consider the extension to the multi-label case, then instead of directly discretizing the simplex, we would have to find a more efficient approach. As we discussed in Section 6 and Supplementary J, we believe the correct approach would be a form of clustering. The complexity of such clustering would depend on N, but the MIFPO would still only depend on the number of clusters. Alternatively, however, one could consider the datapoints themselves (mapped to the simplex), as individual clusters.
> That would result in a problem complexity of $O(N^2)$. Although we do not advocate for such an approach, we note that some recent work (Xian et al 2023) does have $O(N^2)$ complexity (see Supplementary J for additional details).
>
>
> Do these considerations clarify the dependence of MIFPO on the data size?
>
>
>
>
> >    The solution of MIFPO relies on the DCCP convex-concave solver, which in general only guarantees convergence to a local minimum. Does solution quality matter for this approach?
>
> Solution quality defnitely matters! However, we have not observed any considerable local minima issues.
>
> We would also like to note that **any** solution that we find is automatically a lower bound on the true Pareto front. That is, the true front can only be better than what we find. Therefore, even if we had local minima issues, the output of MIFPO would be of interest if its better than other methods.
>
>
>
>
> > The reported empirical study is limited to three variants of a socioeconomic task. There is no assessment on other sensitive features like race and gender or on higher-dimensional data like text
>
> Our experimental protocol follows that of
> (Jovanovic et al 2023), since their method, FARE, is state of the art among methods that learn representations.
> The Health dataset has age (higher or lower than 60) as the sensitive attribute, and the target is a certain co-morbidity rate. The ACSIncome, on the other hand, are socioeconomic datasets, where the label is yearly income (higher or lower than 50k), and the sensitive attribute is gender.
>
>
> Regarding text/image data: We can easily evaluate MIFPO on a number of image and text datasets. As discussed above,  this amounts only to training a standard classifier on the new datset and converting it to probability estimator by calibration. From there, MIFPO is exactly the same as in the other datasets.
>
> However, because none of the methods *we already compare against* has been previously evaluated on such datasets, it is a considerable amount of work to see wether *their existing code* may be adapted. This is not something we can execute in the timeline of the rebuttal, and we hope this is understandable. Please let us know if you believe using such large datasets is important, and we would add them in the final version of the paper.

---

> > ### Comment · Reviewer_DJzQ · 2025-08-04
> >
> > Thank you to the authors for the thoughtful rebuttal. Please feel free to incorporate that stability result as to the number of bins L into the paper. My main concerns have been addressed.

---

### Official Review · Reviewer_8C6m · 2025-06-29

**Clarity:** 3
**Significance:** 3
**Originality:** 4
**Rating:** 5
**Confidence:** 2

**Summary:**

This paper proposes an efficient, model-independent method to compute the fairness-performance Pareto front without requiring the training of complex fair representation models. The authors leverage theoretical analysis to reduce the Pareto computation into a compact discrete optimization problem (MIFPO), which can be solved using disciplined convex-concave programming (DCCP). The method supports arbitrary concave performance measures (beyond accuracy) and can be applied to fair classification directly. Experimental evaluations on benchmark fairness datasets (Health, ACSIncome-CA, ACSIncome-US) demonstrate that MIFPO outperforms in most cases state-of-the-art fair representation and classification methods.

**Questions:**

- The authors state "...this size does not depend directly neither on the feature dimension d, nor 359 on the sample size N and thus the problem scales well in that sense...". Then, could the authors clarify any practical bottlenecks when the feature space is high-dimensional and how feature preprocessing or dimensionality reduction could be (if needed) integrated with MIFPO?

- While the paper effectively computes the fairness-performance Pareto front, it would be valuable to include a qualitative interpretability analysis of the learned trade-offs for downstream decision support. For example, in the Health dataset, what does it mean in practice to choose a point with greater fairness but lower accuracy? Answering this could help practitioners decide which point on the Pareto frontier aligns with their operational goals.

- Since MIFPO relies on the Bayes optimal classifier estimation, could the authors comment on how sensitive the results are to imperfect calibration or estimator mis-specification in practice?

Minor Suggestion:
Consider defining “MIFPO” explicitly in the abstract.

Very nice work !

**Ethical Concerns:**

["NO or VERY MINOR ethics concerns only"]

**Final Justification:**

The authors have addressed most of my concerns with detailed clarifications.
I find the core contributions of the paper to be novel, theoretically grounded, and practically valuable. The methodology is broadly applicable and avoids the need for complex model training, which is a notable strength.

**Limitations:**

Yes. The authors explicitly discuss limitations regarding binary attributes/targets and outline potential extensions while noting the scalability in terms of feature dimensionality and dataset size.

**Paper Formatting Concerns:**

No. All good.

**Quality:**

3

**Strengths And Weaknesses:**

Strengths:
- Introduces theoretically grounded structural results (factorization, invertibility) for fair representations.
- MIFPO addresses the challenge of obtaining the Pareto front efficiently and handles arbitrary concave performance measures, not limited to accuracy.
- The paper provides clear reproducibility details, well-structured algorithmic steps, and evaluation with multiple benchmark datasets.

Weaknesses:
- Evaluation is limited to binary sensitive attributes and binary targets, restricting generality.
- Lack of qualitative interpretability analysis on the learned trade-offs for downstream decision support.
- The implementation of DCCP-based solvers may still encounter local minima in some scenarios (though authors state this was not observed).

---

> ### Author Rebuttal · Authors · 2025-07-31
>
> Thank you for the favourable review!
>
>
> In particular, thanks so much for mentioning that the work is very nice! We were also glad to hear that the results were found to be theoretically grounded and  the presentation and reproducibility to be
> clear.
>
>
> In what follows we discuss the questions and other points raised in the review. Please let us know if there are other questions!
>
>
>
> >Since MIFPO relies on the Bayes optimal classifier estimation, could the authors comment on how sensitive the results are to imperfect calibration or estimator mis-specification in practice?
>
>
> Before discussing our empirical observations, we would like to mention two points:
>
> (a) Calibration, i.e., the estimation of probabilities, can be done quite easily *with guarantees*, given a continuous valued classifer,
> since its esentially a 1d problem. (That may come at some expense of accuracy, however.) See for instance Kumar et al 2019, Berta et al 2024 refs in the paper.
>
> (b) Given such calibration, the Pareto front we find is a lower bound on the true Pareto front. That is, the true front can only be better than what MIFPO computes.
>
>
> In that sense, miss-specification is never too bad, as we err on the safe side. That said, we by all means agree that if the source classifier is poor, then the calibration and the resulting MIFPO Pareto front would be weaker than the true ones.
>
>
> Regarding the empirical results, we have not observed considerable issues with calibration.  On some low dimensional discreet feature datasets, it may be possible to construct "true" probability estimators simply by counting. However, such estimators yielded negligible (less then 0.5 percent) improvements over a generic claibrated xgboost estimator.
>
>
>
>
>
> >Then, could the authors clarify any practical bottlenecks when the feature space is high-dimensional and how feature preprocessing or dimensionality reduction could be (if needed) integrated with MIFPO?
>
>
> In our approach (and also in several other leading fairness classification approaches; see discussion around lines 120-127), the relation between the label and featires is encapsulated by the probability function $\mathbb{P}(Y|X,A)$. Once this function is reliably estimated, it no longer matters for MIFPO what dimension the data was.
>
>
> Of course, when the feature dimension $d$ is high, then estimating $\mathbb{P}(Y|X,A)$ may be non-trivial, depending on the data, and it is here that the dimension reduction may be used.
>
> We note that typically probability estimation can be done quite well given a reliable classifier (see also the related notes in the discussion of misspecificaton above).
>
>
>
>
> >    While the paper effectively computes the fairness-performance Pareto front, it would be valuable to include a qualitative interpretability analysis of the learned trade-offs for downstream decision support. For example, in the Health dataset, what does it mean in practice to choose a point with greater fairness but lower accuracy? Answering this could help practitioners decide which point on the Pareto frontier aligns with their operational goals.
>
> We completely agree that interpretability analysis is important for decision making in practice.
>
> On the ASCIncome dataset [5], as an example, the sensitive attribute is geneder and the label is high/low income.
>
>
> Statistical parity here means that we want a classifier such that the proportion of individuals with predicted high income to be equal between the genders.
>
>
> Note that this is not the situation for the data itself, i.e., the data is considered *biased* in that sense.
> If the classifier was perfect, it would have to flip some income values from their values in data ("true data values"), to become statistical-parity fair. It thus must
> decrease performance to become fair, but this is viewed as corrective bias and is desirable.
>
>
> On the other hand, when the classifier is not perfect to begin with, there may be many classifiers with the same error, some of which are more fair, and some of which are less. In that situation, we would like to pick the most fair one.
>
>
> So the tradeoff question may be also viewed as: Given and amount of "true data values" that we can flip, what is fairest  classifier we can pick?
>
>
> With this in mind, we now look at Figure 3 (the rightmost panel): We see that the best accuracy, without regard of fairness, is around 0.825. Perhaps somewhat strikingly, we also see that we can almost achive this level of accuracy (0.81), with 0 statistical parity distance, i.e., perfect fairness.  MIFPO achieves this, while other methods have considerably larger errors at this fairness level and could errorneously suggest the fairness would result in more information loss.
>
>
> Do these comments help clarify the interpretability question?
>
> While here we considered a brief example, for a general in-depth analysis of the notions of fairness tradeoffs, we refer to the excellent book [6].
>
>
> Finally, we note that while the Health dataset [1], originally mentioned in the review, is a standard banchmark in fairness, see for instance [2], [3], [4], and is a very good example for testing *computational aspects of algorithms*, we feel we do not have the appropriate background to discuss the
> *interpretation* for this particular dataset.
>
>
>
>
> [1] Kaggle. Health heritage prize, 2012. URL https://www.kaggle.com/c/hhp
>
> [2] Jovanovic et. al. ICML 2023,  Fare: Provably fair representation learning with practical certificates.
>
> [3] Louizos et al, ICLR 2016, The variational fair autoencoder.
>
> [4] Madras et al. NeurIPS 2018, Predict responsibly: Improving fairness and accuracy by learning to defer.
>
> [5] Ding et al, NeurIPS 2021 Retiring
> adult: New datasets for fair machine learning. Advances
> in Neural Information Processing Systems, 34, 2021
>
> [6] Barocas, S. et al (2023). Fairness and machine learning: Limitations and opportunities.

---

> > ### Comment · Reviewer_8C6m · 2025-08-06
> >
> > Dear authors,
> >
> >
> > Thank you for your response and clarifications, which have addressed most of my concerns.
> >
> >
> > I'll upgrade my score to Accept.

---

### Official Review · Reviewer_rVkf · 2025-06-30

**Clarity:** 2
**Significance:** 2
**Originality:** 3
**Rating:** 5
**Confidence:** 3

**Summary:**

The paper creates a method to get the pareto-front of a representation of samples in the dataset with regards to statistical parity (a fairness measure) and performance of the method. In order to create this pareto-front mathematical proofs are provided, showing a theoretical basis for the information that can still be captured in the representation while becoming independent of a binary sensitive attribute.

**Questions:**

I would like the authors to provide some clarity on how their method will be made accessible to the research field itself in the form of a technical implementation. Given a sufficient answer to this question, I would improve my score.

**Ethical Concerns:**

["NO or VERY MINOR ethics concerns only"]

**Final Justification:**

The authors addressed my concern for the need of a code implementation to accompany the paper in order to provide a significant contribution for the research field.

The possibility of having pareto-fronts on datasets to conduct fairness experiments is useful for determining method capabilities. The paper achieves this goal from a theoretical point and provides experimental results showcasing its use.

**Limitations:**

yes

**Paper Formatting Concerns:**

The authors used an incorrect citation style.

**Quality:**

3

**Strengths And Weaknesses:**

The paper makes careful work of providing mathematical proofs for the steps that lead to creating the perato-front;

The authors make careful work of providing a detailed related works section, allowing the reader to place the paper in the context of the research field.

The authors accompany their theoretical work with some experimental works. These results do show that the pareto-front is not fully a pareto-front as some of the results from the models exhibit better performance. Related to this, the authors state that the other methods are run several times for different hyperparameters. It would have been good to run each combination of hyperparameters for several seeds in order to report the volatility of the resulting method, perhaps with the use of confidence ellipses.

Although this is largely a theoretical work on fairness, it would have been nice of some reference was made that this pareto-front only works on the dataset and that it's performance in the real-world might still be different as there are biases in the data that a method cannot account for.

The most significant results of this paper is the possibility of creating pareto-fronts for datasets. While the theoretical proofs are interesting and necessary, its contribution would be the algorithm to generate the pareto-front. Currently no implementation of this code is provided with the submission (no link in the paper itself to the code, nor the code itself included in the supplementary material).

The scope of the paper is rather limited as the method only works for binary sensitive attributes and binary prediction. The authors do note future work in order to broaden the scope for these purposes too.

---

> ### Author Rebuttal · Authors · 2025-07-31
>
> Thank you for your feedback!
>
> We were glad to hear that the reviewer found our proofs to be interesting and careful, and that the results are well positioned  within the general context of the research field.
>
>
> In what follows, we discuss the other points raised in this review. Please let us know if there are any further questions, we would be glad continue the discussion!
>
>
> First, to address the main question in the review:
> > I would like the authors to provide some clarity on how their method will be made accessible to the research field itself in the form of a technical implementation. Given a sufficient answer to this question, I would improve my score.
>
>
> We completely agree with the reviewer on the importance of making the imlementation publicly accessible.
>  We are fully committed to ensuring our work is accessible and useful to the research community, and we have already prepared the code repository for public release. We are also committed to this as part of our research obligations under our funding grant agreement.
>
> While the conference's current policy prevents us from sharing a link, we can provide clarity on the technical implementation of our software package, which is ready for publication:
> 1. Our code is a Python package, installable via `pip`. It follows Scikit-learn's API guidelines(as scikit-learn estimator) for seamless use.
> 2. For tabular data, the API requires a single table `X`, a labels vector `y`, and the column name of the sensitive attribute. In this mode, users can optionally provide their own pre-trained calibrated classifiers. If no classifiers are given, the package defaults to using an internally calibrated XGBoost model to learn the probabilities $P(Y|X,S)$.
> 3. For non-tabular data (e.g., images, text), the API accepts data that is already pre-split by the sensitive attribute (`X_0`, `X_1`, `y_0`, `y_1`). In this scenario, the user must also provide the corresponding pre-trained calibrated classifiers capable of handling the specific data type.
> 4. The package and the paper's experiments are in **separate repositories**. The experiments repository uses the package as a third-party dependency, which demonstrates its real-world usability and provides a clear usage example.
>
>
> Do these  notes address the question  regarding the publishing of the code?
>
>
> >The authors state that the other methods are run several times for different hyperparameters. It would have been good to run each combination of hyperparameters for several seeds in order to report the volatility of the resulting method, perhaps with the use of confidence ellipses.
>
> We assume that the discussion here is about the runs of the existing methods we compare with. There are over 200 different runs, that correspond to different methods and different hyperparameter values (that result in the different tradeoff points for those methods). Thus it will be quite time consuming to repeat everything with multiple seeds. Moreover, since every separate run uses a separate seed, while individual run results might shift, it is quite unlikely that the whole *tradeoff line* of the method would shift significantly (as it contains multiple independet runs). Thus we belive that the presented results are fairly stable.
> However, if the reviewer finds this experiment to be of high imprtance, please let us known, and we will include this in the final version of the paper.
>
>
> For our method, MIFPO, we have observed very little dependence on the initailization parameters (of the probabiliy estimator, and of the DCCP initialization). The variance clouds would be practically invisible on the plots. We will include these numbers formally in the supplementary.
>
>
>
> >Although this is largely a theoretical work on fairness, it would have been nice of some reference was made that this pareto-front only works on the dataset and that it's performance in the real-world might still be different as there are biases in the data that a method cannot account for.
>
> Unfortunately, we are not quite sure what was meant in this point. Could the reviewer clarify? We would be glad to discuss this further.
>
> In general, we follow the same experimental protocol as other works in the field. We agree that of course, there may be some aspects of bias that are not captured by any given protocol.
>
>
> >The scope of the paper is rather limited as the method only works for binary sensitive attributes and binary prediction. The authors do note future work in order to broaden the scope for these purposes too.
>
> Indeed, for non-binary sensitive attributes and labels, we discuss the associated difficulties, and considerations towards resolving them, in the Conclusions and Future Work section.
>
> We would like to note, however,  that non-binary quantities appear to be an issue *for any method* for which computational complexity can be meaningfully quantified, not just ours.
> For instance, the complexity of Xian et al, 2023, for multi-valued labels would be similar to the complexity of our method (See Supplementary J), despite the fact that they solve a simpler problem (only accuracy metric, only classification). Additional details and discussion on this topic can be found in Section 2 and Supplementary J.

---

> > ### Comment · Reviewer_rVkf · 2025-08-04
> >
> > I would like to thank the authors for their rebuttal and I am glad to see that a thorough plan is in place to make this code for plotting the pareto-fronts accessible in the form of a pip package. I hope to see a reference to the package added in the camera-ready version of the paper. I have changed my score of the paper in response of the rebuttal provided by the authors.
> >
> > The comment on running the experiments for several seeds and reporting the average with a confidence ellips was more a suggestion for the authors for future works. The results in the paper are sufficiently interpretable and I do not expect additional experiments to be run. In future works the approach of multiple seeds would simply strengthen results in these type of experiments.
> >
> > The comment that the pareto-front is theoretical was referencing that this pareto-front is for the reported labels. Label bias is a common problem in fairness datasets and therefore the pareto-front might not be the same for the actual ground truth. This is not something that can be tackled in this work, but could be an interesting remark to add in the paper stating an inherent limitation of some fairness datasets.
> >
> > The comment on the limited scope was not meant to be critical for the paper, but rather included for the completeness of the review.

---

### Note · Authors · 2025-08-15

We would like to take this opportunity to thank the reviewers again for the helpful reviews. Many Thanks!




*@Reviewer rVkf:* Thanks so much for your comments and for updating the score! The software package will be made public on github and the reference to it  will be added to the final version of the paper.


*@Reviewer 8C6m:* Many thanks for the comments and for raising the score! We were glad to hear that the concerns in the review were addressed. We will add the discussions on interpretability, and on the dependence on dimension and miss classification to  the final version of the paper.




*@Reviewer DJzQ:* Thanks again for your comments! We were glad to hear that the main concerns of the review were addressed. We will add to the paper the stability result discussed in our original response, relating closeness of distributions on the simplex to the closeness of the Pareto fronts, and which in turn allows to quantify the dependence on L.

---

### Decision · Program_Chairs · 2025-09-17

**Decision:**

Accept (spotlight)

**Comment:**

The authors characterize the possibility of fair representation learning in single-group binary classification settings. They show that the computation of the fairness-utility pareto front reduces to a discrete optimization problem, for which existing methods can be deployed. Reviewers were positive about the paper, and expressed an excitement for the future work opened up by a theoretically-grounded model-agnostic method to generate pareto fronts for fairness datasets. While some limitations (e.g. restriction to binary targets) were acknowledged, I recommend accepting the paper and encourage the authors to integrate reviewer feedback (e.g. publishing the github repo) into the final revision.